# Robust optimal design of FOPID controller for five bar linkage robot in a Cyber-Physical System: A new simulation-optimization approach

Amir Parnianifard[1]☉*, Ali Zemouche[2‡], Ratchatin Chancharoen[3‡], Muhammad Ali Imran[4‡], Lunchakorn Wuttisittikulkij[1]☉*

1 Wireless Communication Ecosystem Research Unit, Department of Electrical Engineering, Faculty of Engineering, Chulalongkorn University, Bangkok, Thailand, 2 Wireless Communication Ecosystem Research Unit, Department of Mechanical Engineering, Faculty of Engineering, Chulalongkorn University, Bangkok, Thailand, 3 University of Lorraine, CRAN UMR CNRS 7039, Cosnes et Romain, France, 4 School of Engineering, University of Glasgow, Glasgow, United Kingdom

☉ These authors contributed equally to this work.
‡ These authors also contributed equally to this work.
* Lunchakorn.W@chula.ac.th (LW); amir.p@chula.ac.th (AP)

**Data Availability Statement:** The supplementary materials including all Matlab® codes and functions and Excel data set and analysis required to

## Abstract

This paper aims to further increase the reliability of optimal results by setting the simulation conditions to be as close as possible to the real or actual operation to create a Cyber-Physical System (CPS) view for the installation of the Fractional-Order PID (FOPID) controller. For this purpose, we consider two different sources of variability in such a CPS control model. The first source refers to the changeability of a target of the control model (multiple setpoints) because of environmental noise factors and the second source refers to an anomaly in sensors that is raised in a feedback loop. We develop a new approach to optimize two objective functions under uncertainty including signal energy control and response error control while obtaining the robustness among the source of variability with the lowest computational cost. A new hybrid surrogate-metaheuristic approach is developed using Particle Swarm Optimization (PSO) to update the Gaussian Process (GP) surrogate for a sequential improvement of the robust optimal result. The application of efficient global optimization is extended to estimate surrogate prediction error with less computational cost using a jackknife leave-one-out estimator. This paper examines the challenges of such a robust multi-objective optimization for FOPID control of a five-bar linkage robot manipulator. The results show the applicability and effectiveness of our proposed method in obtaining robustness and reliability in a CPS control system by tackling required computational efforts.

## 1. Introduction

Nowadays, developing processes in the engineering world is strongly associated with computer simulations. These computer codes can collect appropriate information about the characteristics

reproducing and replicating of our study's findings have been shared publicly in the Zenodo repository via DOI and URL as below: DOI: 10.5281/zenodo. 4266126 URL: https://doi.org/10.5281/zenodo. 4266126.

**Funding:** This research project is supported by Second Century Fund (C2F), Chulalongkorn University, Bangkok. The authors acknowledge the support by Second Century Fund (C2F), Chulalongkorn University, Bangkok for funding and postdoctoral fellowship to Amir Parnianifard (first and corresponding author). The funders had no role in study design, data collection and analysis, decision to publish, or preparation of the manuscript.

**Competing interests:** The authors have declared that no competing interests exist.

of engineering problems before actually running the process. Computer simulations can provide a rapid investigation of various alternative designs to decrease the required time to improve the system. In addition, most numerical analyses for engineering problems make a well-suited use of mathematical programming. The main goals of simulation include what-if study of a model or sensitivity analysis and optimization and validation of the model [1]. The essential benefit of simulation is its ability to cover complex processes, either deterministic or random while eliminating mathematical sophistication [2]. Clearly, because of the complexity of mathematical formulation analyzing in many real-world optimization problems, simulation-optimization methods become necessary to find more interest and popularity than other optimization methods [3–5].

Cyber-Physical System (CPS) combines physical objects or systems with integrated computational facilities and data storage [6]. CPS is a key enabling technology in systems intelligence. In CPS embedded computers and networks, the physical processes are controlled usually with feedback loops where physical processes affect computations and vice versa [7–10]. CPS is a multidimensional and complex system that integrates the cyber world with the dynamic physical world. Integrating physical processes with computer systems is the main challenge presented in CPS as the computational cyber part continuously senses the state of the physical system and applies decisions and actions for its control [11]. The integration and collaboration of three terms including computing, communication, and control are known as "3C" [12, 13], CPS provides sensing, real-time optimization, information feedback, dynamic control, and other services, see Fig 1. In recent years, the application of CPS has been widely considered in different fields such as aerospace [14–16], defense [17, 18], energy systems [19, 20], healthcare [21–24], vehicle [25–27], and others [28–30].

In industrial practice, many CPS systems have been designed by decoupling the control system design. In this way, CPS and real-time interaction are achieved in order to monitor and control physical entities in a reliable, safe, collaborative, robust, and efficient way [12, 13]. Using precise calculations to control a seemingly unpredictable physical environment is a great challenge [31]. After the CPS control system is designed and modeled by extensive simulation, tuning methods need to be expanded to address uncertainty and random disturbances in the system. In addition, ignoring the impact of uncertainty on the optimization model, the obtained optimal results may be far from the true optimum settings [32]. One of the main features in a reliable CPS design is the stability feature (robustness), which means no matter how the environment generates noise and uncertain factors, the control system should always reach a stable decision result eventually [33]. Robustness in the CPS control system seeks to achieve a certain level of performance with possible modeling errors in the forms of parametric or nonparametric uncertainties [34]. However, considering uncertainty and random disturbances, while keeping the function and operation of the system, has been computationally time-consuming and costly.

Because of uncertainty, more complexity in the real-time control implementation of CPS is unavoidable. So, looking for less expensive computational methods of optimization considering uncertainties has become interesting among most engineering applications [35]. To overcome such computational difficulties, researchers have applied surrogate-based learning methods (e.g. polynomial regression, GP, and radial basis function) [36–39]. Surrogate-based methods can 'learn' the problem behaviors and approximate the function value. These approximation models can accelerate the function evaluation as well as the estimation of the function value with acceptable accuracy. Also, they can improve the optimization performance and provide a better final solution. Various types of real-world engineering optimization problems have been developed by applying surrogate-based methods. These optimization problems include dynamic and stochastic control system design, sub-communities in machine learning

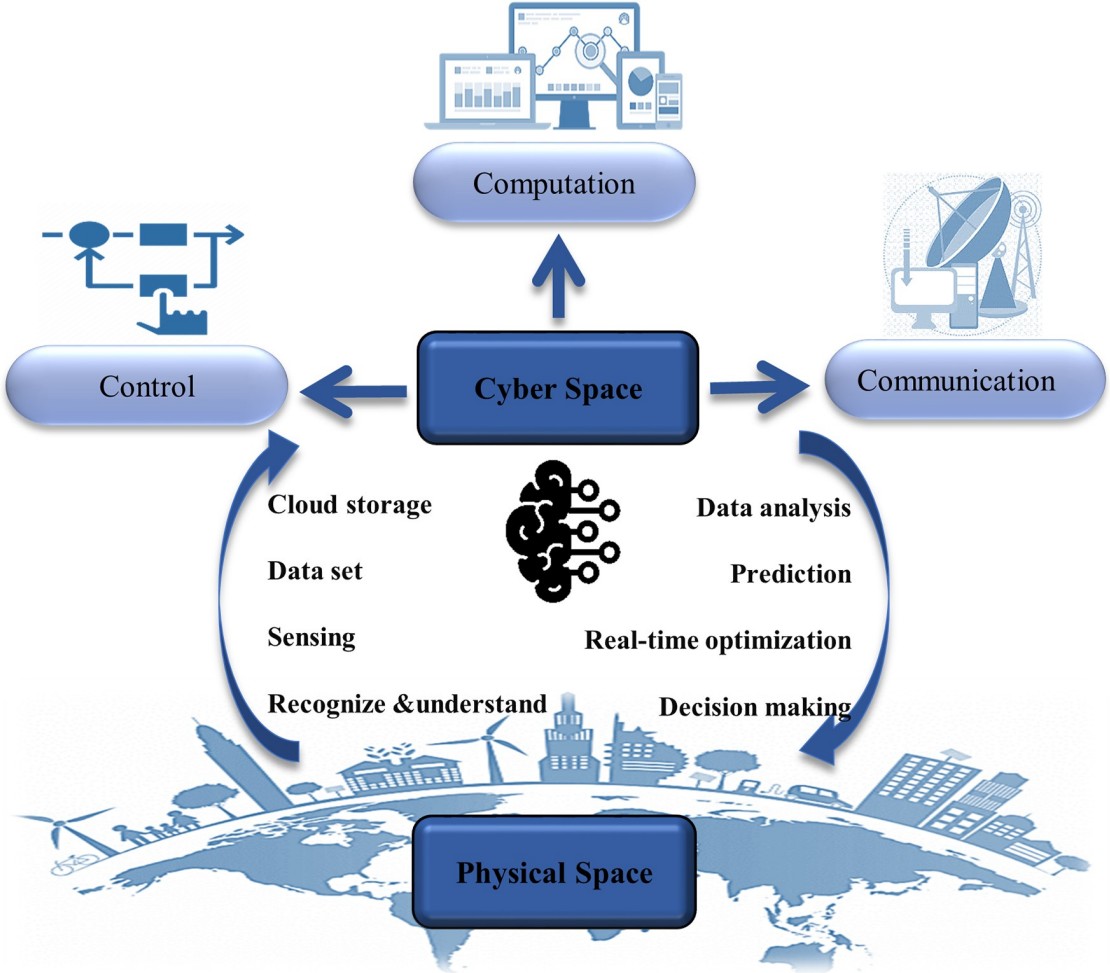

**Fig 1. Overall representation of Cyber-Physical System (CPS).**

problems, discrete event systems (e.g. queues, operations, and networks), manufacturing, medicine and biology, engineering, computer science, electronics, transportation, and logistics, see [3, 5, 38, 40–42]. However, several studies have systematically illustrated the applications of surrogate-based optimization algorithms [38, 39, 43–45].

In this paper, a new outline of robust real-time optimization in the CPS control model under the effect of environmental factors (also known as noise factors or uncertainty, see [4, 5]), and variability in feedback loop due to sensor's anomaly is studied. The main contributions of this study are as follows:

1. In this paper, we propose a new CPS framework of the control system for a five-bar linkage robot manipulator by considering the effect of uncertainty (sources of variability) in the stochastic model. The first source is related to a real-time setpoint that is predicted by learning from collected data (e.g. surrogate) over CPS environmental factors and the second variability is found in output's feedback due to anomaly in sensors. Besides, energy consumption and response error are optimized as a robust multi-objective optimization model by Pareto frontier estimation in the real-time computational part of the CPS model.

2. A new hybrid surrogate/metaheuristic algorithm for robust tuning of FOPID controller in the stochastic control system is proposed. The proposed hybrid GP/PSO algorithm has the advantages of both GP surrogates in learning the behavior of the model in an efficient global optimization with PSO metaheuristic in convergence searching for optimum results. We apply the straightforward jackknife leave-one-out technique to estimate surrogate prediction error applied in efficient global optimization.

3. The proposed algorithm can analyze the sensitivity of the obtained optimal results in such stochastic environments using the same collected data obtained among optimization procedure and simulation doesn't need to be run anymore for computing the confidence intervals of robust optimal results (i.e. this algorithm does not to increase the number of function evaluations for sensitivity analysis).

The rest of this paper is organized as follows. Section 2 provides more details about real-time FOPID control when two types of uncertainties (noises) including environmental factors and sensor anomaly are considered in a CPS framework. Materials and methods of the proposed algorithm to handle robust multi-objective optimization of a CPS control system are elaborated in Section 3. In Section 4, the applicability and effectiveness of the proposed approach are examined to provide robustness and reliability in the robust optimal design of the FOPID controller in the CPS framework of a five-bar linkage robot manipulator. Finally, this paper is concluded in Section 6.

## 2. Point of view

The existing uncertainties and anomalies in the cyber environment have resulted in emerging concerns about the traditional control system [34]. In real-time control of CPS, physical process variables are monitored and processed by intelligent controllers for keeping the values of safety parameters between the given thresholds. Environmental conditions can affect system dynamics and also the controller function [9]. The precision of computing must interface with the uncertainty and the noise in the physical environment [46]. The physical world, however, is not entirely predictable. Normally, the CPS does not operate in a controlled environment. So, it must be robust to uncertainty (unexpected conditions) and adaptable to subsystem failures [8].

### 2.1 Nomenclature

The main parameters and symbols used in the proposed algorithm are revealed in Table 1.

### 2.2 FOPID controller

In this paper, for better control, a fractional-order $PI^\lambda D^\mu$ controller is used. Currently, fractional-order controllers are being extensively used by many scientists to achieve the most robust performance of the systems [47]. The main reason for choosing FOPID controllers is their additional degrees of freedom that result in a better control performance [48, 49]. A generalized FOPID controller was first introduced by [50] which proposed $PI^\lambda D^\mu$ controller involving a $\lambda$ order integer and a $\mu$ order differentiator. The differential equation of a fractional-order $PI^\lambda D^\mu$ controller is defined by:

$$u(t) = K_p e(t) + K_i D_t^{-\lambda} e(t) + K_d D_t^\mu e(t) \tag{1}$$

**Table 1. The table of nomenclature.**

| Notation | Description |
|---|---|
| $K_i, K_p, K_d,$ $\lambda, \mu$ | FOPID gain parameters (decision variables in an optimization model in this study). |
| $e(t)$ | The error of the control system in the time moment $t$ (distance between the output of the system with the desired set point in time moment $t$). |
| $\widehat{s}(t)$ | Desired setpoint in the control system (first uncertain variable in this study). |
| $\tilde{\alpha}$ | Percent of variability in the feedback loop of the control system (second uncertain variable in this study). |
| $L_s, U_s$ | The lower and upper limit for the uncertain variable $\widehat{s}(t)$. |
| $L_\alpha, U_\alpha$ | The lower and upper limit for the uncertain variable $\tilde{\alpha}$. |
| $y(t)$ | The output of the plant in a control system in the time moment $t$. |
| SEC | Signal Energy Control |
| REC | Response Error Control |
| $F_1, F_2, OF$ | First, second, and overall objective functions in the optimization model respectively. |
| $\theta$ | A user-defined weighting factor used in overall function formulation and shows the tendency of the model toward $F_1$ or $F_2$ functions. |
| $Mean_s, Std_s$ | Mean and standard deviation of $s$th input combination. Regarding the crossed array design, these statistical parameters are computed through repetitions of $s$th input combination over different uncertainty scenarios. |
| $SNR_s$ | Signal to noise ratio of $s$th input combination and computed by $SNR_s = 10\log[Mean_s\,2 + \omega * Std_s\,2]$. |
| $\omega$ | A weighting parameter that is introduced to allow for individual emphasis on the minimization of variations in $SNR_s$ formulation. |
| $l \times m$ | The number of simulation experiments regarding the structure of crossed array design with $l$ input combinations and $m$ uncertainty scenarios. |
| $EI(c)$ | The expected improvement that can be considered for the candidate point $c$ from the best point so far. |
| $\gamma$ | Type I error and shows the probability of becoming infeasible from estimated confidence intervals. |
| $R_{p,s}$ | Performance measure criteria |
| CIs | Confidence intervals for optimal result using the augmented bootstrapping technique (employ the same set of data used for optimization procedure). |
| $\widehat{S}_c$ | GP surrogate prediction error used in expected improvement $EI(c)$ formulation. In this paper, the surrogate prediction error computed by the Jackknife leave-one-out approach. |

where $e(t)$ calculates an error value as the difference between a desired setpoint and a measured process output in the time moment $t$. The controller attempts to minimize the error over time by adjustment of a control variable $u(t)$. The reliability of the FOPID controller depends on the optimal design of three gain parameters ($K_i, K_p, K_d$) and two order parameters ($\lambda, \mu$). However, we try to further increase the reliability of the tuning result by setting the traits of the simulation model to be as close as possible to the practical condition to make a CPS outline for the FOPID controller. The FOPID control system with a single setpoint does not express the aspects of the behavior that are essential to the system in the context of CPS. Moreover, we challenge the robust control to achieve CPS stability when the uncertainty in environmental conditions is the source of variability of the setpoints in the control system. In addition, an uncertain anomaly in sensors causes the noise (variability) in the control feedback loop. Moreover, we aim to tune the FOPID controller robustly in such a CPS control system with real-time setpoints and noise in the model's feedback. Fig 2 shows the control outline of CPS with real-time setpoints and noise in the model's feedback. The application of the integer-order and fractional-order of the PID controller in CPSs has been studied in [49, 51–53].

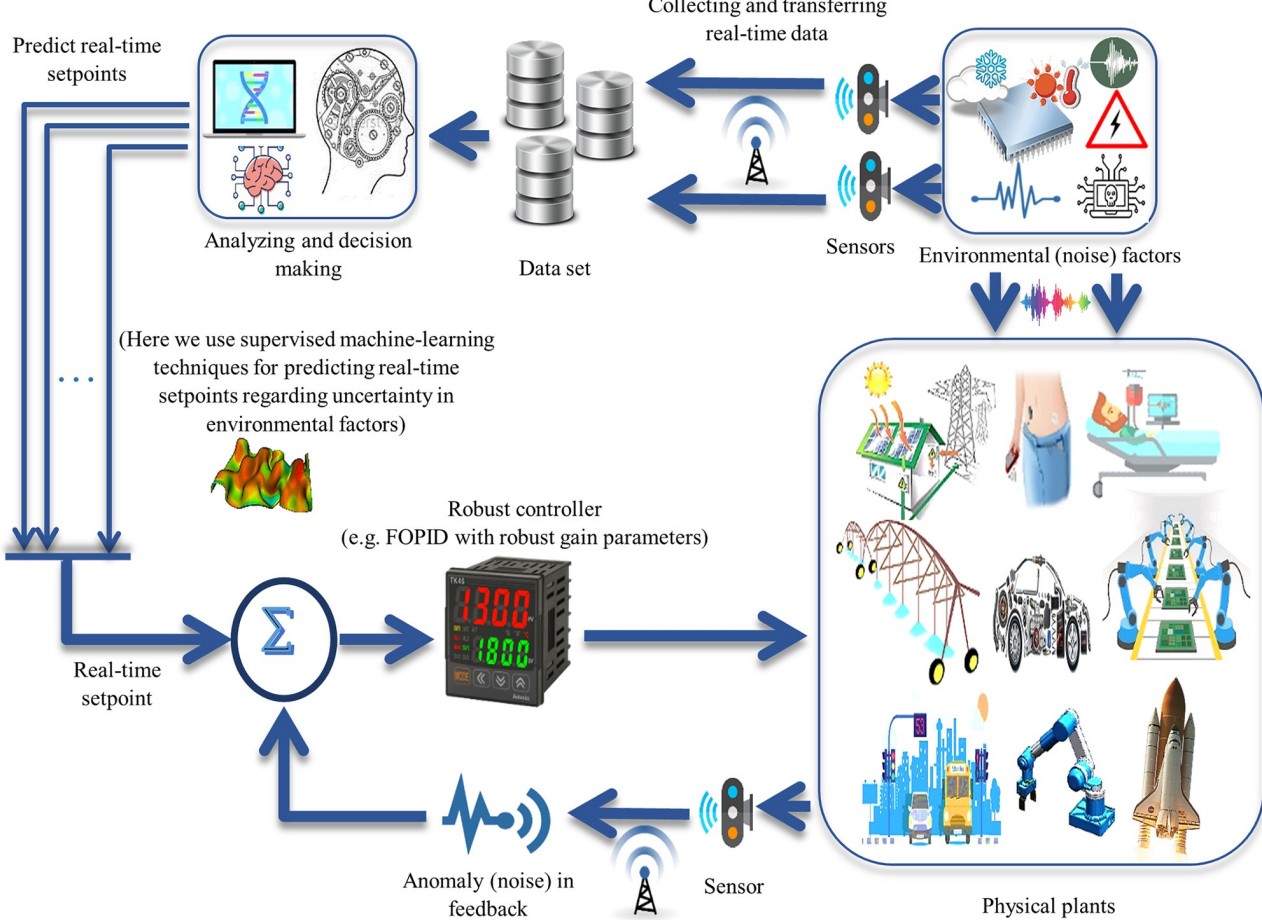

**Fig 2. The control framework of CPS with real-time setpoints and noise in model's feedback.** The environmental factors would be predicted and applied as a real-time setpoint and anomaly in sensor is estimated in feedback loop. Gain parameters and order parameters in FOPID controller are tuned to be robust against source of variability.

## 2.3 Uncertainty in the CPS control model

Assume $\tilde{z}_1(t), \tilde{z}_2(t), \ldots, \tilde{z}_n(t)$ are the environmental (uncertain) factors in such a CPS control outline. It should be noted that a real-time setpoint of the control system at time moment $t$ is affected by variability on the environmental (uncertain) factors. Furthermore, the decision policy needs to be able to predict real-time setpoints regarding the data collected from the uncertain environmental factors so far. Here, we use supervised learning of data collected so far from the environment (e.g. polynomial regression function, $\widehat{f}[\tilde{z}_1(t), \tilde{z}_2(t), \ldots, \tilde{z}_n(t)]$) and predict the real-time setpoint $\widehat{s}(t) = \frac{\widehat{df}}{dt}, \widehat{s}(t) \in [L_s, U_s]$ in the control system. In addition, an anomaly in the sensor to convey response feedback is assumed as uncertainty that causes the variability in the tuning of the FOPID controller. Assume that the true response of model $y(t)$ is varied by $\tilde{\alpha}\%$ where $\tilde{\alpha}$ is uncertain variable ($\tilde{\alpha} \in [L_\alpha, U_\alpha]$), thus $\tilde{y}(t) = y(t) \times (1 + \tilde{\alpha})$ is a true response that is transmitted to the controller at time-step $t$. Fig 3 shows a block diagram representation of the CPS control system by considering both types of uncertainty including environmental factors, and sensor anomaly.

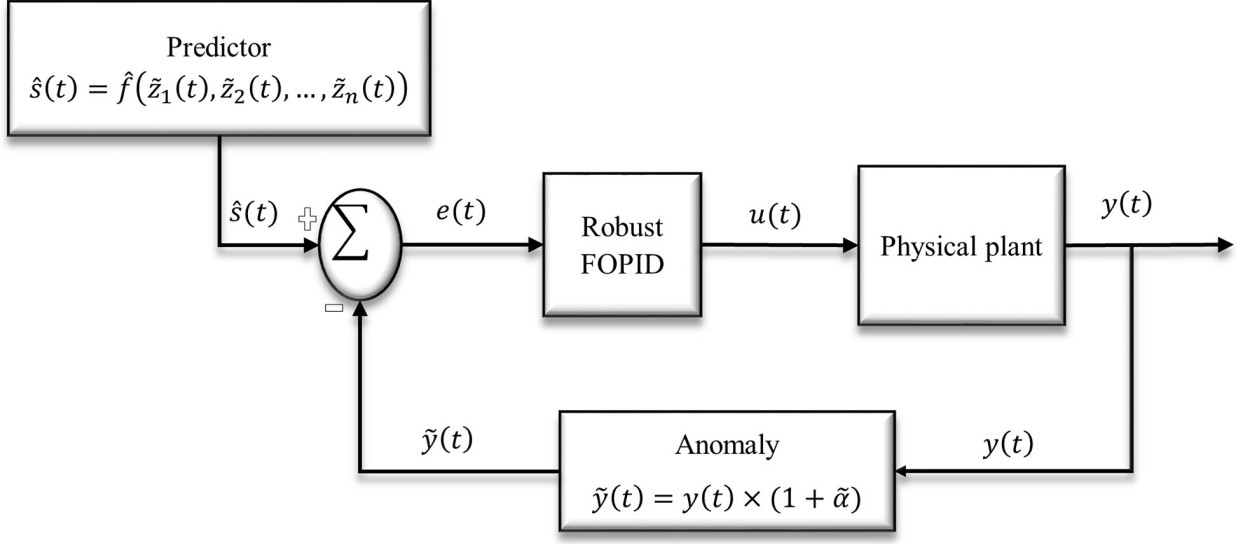

**Fig 3. The block diagram of robust FOPID control in CPS framework with real-time setpoints and noise in model's feedback.** Real-time setpoint is estimated by approximation function of environmental factors $(\tilde{z}_1(t), \tilde{z}_2(t), \ldots, \tilde{z}_n(t))$. Anomaly in sensor's feedback is function of uncertain variable $\tilde{\alpha}$. FOPID gain parameters and order parameters are tuned robustly in such a way to make CPS insensitive against sources of variability in system.

### 2.4 Objective functions

This study aims at optimizing a robust multi-objective model of the FOPID tuning in the CPS framework by considering two different objective functions (e.g. performance criteria). The authors in [54] considered the amount of energy consumed to control the plant. They applied this measure to compare the optimal results obtained by different methods. However, they don't use the energy consumption factor for the tuning procedure. Here, we define the Signal Energy Control (SEC) as the first objective function to optimize the energy that is consumed in the time domain $0 \leq t \leq T$:

$$F_1 = \frac{\log(SEC + 1)}{M_1} \tag{2}$$

and

$$SEC = \int_0^T |u(t)| dt = \int_0^T |K_p e(t) + K_i D_t^{-\lambda} e(t) + K_d D_t^{\mu} e(t)| dt \tag{3}$$

where $M_1$ is a big value that is defined by user and is used to normalize the first objective function in [0,1], so that $M_1 > \log(\max_{0 \leq t \leq T} SEC)$.

We define the second objective function, namely Response Error Control (REC) inspired by common integral absolute error criteria [47, 48] as below:

$$F_2 = \frac{\log(REC + 1)}{M_2} \tag{4}$$

and

$$REC = \int_0^T |e(t)| dt = \int_0^T |\tilde{y}(t) - \hat{s}(t)| dt \tag{5}$$

where $M_2$ shows a big value that is defined by user and is used to normalize the second objective function in [0,1], so that $M_2 > \log(\max_{0 \leq t \leq T} REC)$. Notably, we use a logarithmic scale for both objective functions to smooth the large differences between the values (i.e. cases in which one or a few points are much larger than the bulk of the data). As mentioned earlier, the real-time setpoint $\hat{s}(t)$ in Eq (5) can be predicted on-time by easy-to-apply supervised learning like polynomial regression as a function of environmental uncertain factor(s).

## 2.5 Overall objective function

To combine both objective functions including the signal energy control (see Eq (2)) with response error control (see Eq (4)) to be used as a single objective model, we apply $Lp$-mertic approach by $p = 2$ (i.e. for more information about $Lp$-mertic approach in multi-objective optimization, refer to [55]). Assume $s = (1,2,\ldots,l)$ is the vector of input combination, then we define the Overall Function (OF) as below:

$$OF = \{\theta(F_1)^2 + (1-\theta)(F_2)^2\}^{\frac{1}{2}} \quad, for(s = 1, 2, \ldots, l) \tag{6}$$

where $\theta$ is a user-defined weighting factor ($0 \leq \theta \leq 1$) that indicates the tendency of the model toward optimization based on each objective function $F_1$ and $F_2$, see Eqs (2) and (4). Fluctuating this magnitude ($\theta$) provides the capture of Pareto frontier (also called Pareto optimal efficiency) to make a trade-off between each objective function. This approach is a classical method to solve optimization problems when the model is faced with multiple criteria [56]. In fact, the set of optimal solutions obtained from fluctuating $\theta$ in [0,1] provides an estimate of the Pareto frontier.

## 3. Proposed algorithm

In this section, we propose a promising technique for optimization under uncertainty using augmented efficient global optimization using the jackknife leave-one-out technique to estimate GP prediction error hybrid GP/PSO method. For this purpose, we first explain the main materials and methods used in the proposed algorithm briefly and then sketch the algorithmic steps in the proposed approach.

### 3.1 Materials and methods

**3.1.1 Gaussian Process (GP) surrogate.** GP which is also known as kriging is a non-parametric Bayesian approach to supervised learning [57]. GP is an interpolation method that can cover deterministic data and is highly flexible due to its ability to employ various ranges of correlation functions [58]. In a GP model, a combination of a polynomial model and the realization of a stationary point are assumed by the form of:

$$y = f(X) + Z(X) + \varepsilon \tag{7}$$

$$f(X) = \sum_{p=0}^{k} \widehat{\beta}_p f_p(X) \tag{8}$$

where the polynomial terms of $f_p(X)$ are typically the first or the second-order response surface approach and coefficients $\widehat{\beta}_p$ are regression parameters ($p = 0,1, \ldots, k$). This type of GP approximation is called the universal GP, while in the ordinary GP, instead of $f(X)$, the constant mean $\mu = E(y(x))$ is used. The term $\varepsilon$ describes the approximation error and the term $Z(X)$ represents the realization of a stochastic process which in general is a normally

distributed Gaussian random process with zero mean, variance $\sigma^2$, and non-zero covariance. The correlation function of $Z(X)$ is defined by:

$$Cov[Z(x_k), Z(x_j)] = \sigma^2 R(x_k, x_j) \tag{9}$$

where $\sigma^2$ is the process variance and $R(x_k, x_j)$ is the correlation function that can be chosen from different correlation functions which were proposed in the literature (e.g. exponential, Gaussian, linear, spherical, cubic, and spline), see [59, 60]. Today, GP surrogate has been used as a widespread global approximation technique that is applied widely in control engineering design problems [40, 61].

**3.1.2 Particle Swarm Optimizer (PSO).** The canonical PSO algorithm was proposed by [62] and is inspired by the social behavior of swarms such as bird flocking or fish schooling. The parameters of PSO consist of the number of particles, position of agent in the solution space, velocity, and neighborhood of particles (communication of topology). The PSO algorithm begins with initializing the population. The second step is to calculate the fitness values of each particle, followed by updating individual and global bests as the third step. Then, velocity and the position of the particles become updated (step four). The second to fourth steps are repeated until the termination condition is satisfied [63, 64]. The PSO algorithm is formulated as follows [62–64]:

$$v_{id}^{t+1} = w.v_{id}^t + c_1.rand(0,1).(p_{id}^t - x_{id}^t) + c_2.rand(0,1).(p_{gd}^t - x_{id}^t)$$
$$\text{and } x_{id}^{t+1} = x_{id}^t + v_{id}^{t+1} \tag{10}$$

where $w$ is the inertia weight factor, $v_{id}^t$ and $x_{id}^t$ are particle velocity and particle position respectively. $d$ is the dimension in the search space, $i$ is the particle index, and $t$ is the iteration number. Expressions $c_1$ and $c_2$ represent the speeds of regulating the length when flying towards the most optimal particles of the whole swarm and the most optimal individual particle. The term $p_i$ is the best position achieved by particle $i$ so far and $p_g$ is the best position found by the neighbors of particle $i$. The expression $rand(0,1)$ shows the random values between 0 and 1. The exploration happens if either or both of the differences between the particle's best $(p_{id}^t)$ and previous particle's position $(x_{id}^t)$, and between the population's all-time best $(p_{gd}^t)$ and previous particle's position $(x_{id}^t)$ are large. In addition, exploitation occurs when these two values are both small. PSO has attracted wide attention in control engineering design problems due to its algorithmic simplicity and powerful search performance [54, 65]. However, PSO algorithm that requires a large number of fitness evaluations before locating the global optimum is often prevented from being applied to computationally expensive real-world problems [66]. Therefore, surrogate-assisted PSO metaheuristic optimization algorithms have been focused in the literature, see [66–68].

**3.1.3 Uncertainty management.** Here, we follow [39, 69, 70] and inspire Taguchi's overview of robust design [71] for dealing with uncertainty as a source of variability in the model. However, we expand Taguchi's robust design terminology and apply its definition for environmental noise factors in such a CPS control system. But in this study, we replace the statistical approach of Taguchi viewpoint with augmented efficient global optimization using jackknife leave-one-out technique and hybrid GP/POS approach. Furthermore, we first intersect two sampling design sets. One sampling design is for decision variables (inner array) and another is for uncertain variables (outer array). Given that $s = (1, 2, . . ., l)$ is the vector of sample points over decision variables, and $r = (1, 2, . . ., m)$ is the vector of uncertainty scenarios, so $l \times m$ input combinations are designed, and the real model (or true simulation model) are evaluated $l \times m$ times to collect relevant simulation outputs, see Fig 4. Assume $Y$ is the $l \times m$ matrix of simulation outputs (i.e. in this study the simulation outputs include OF values that can be

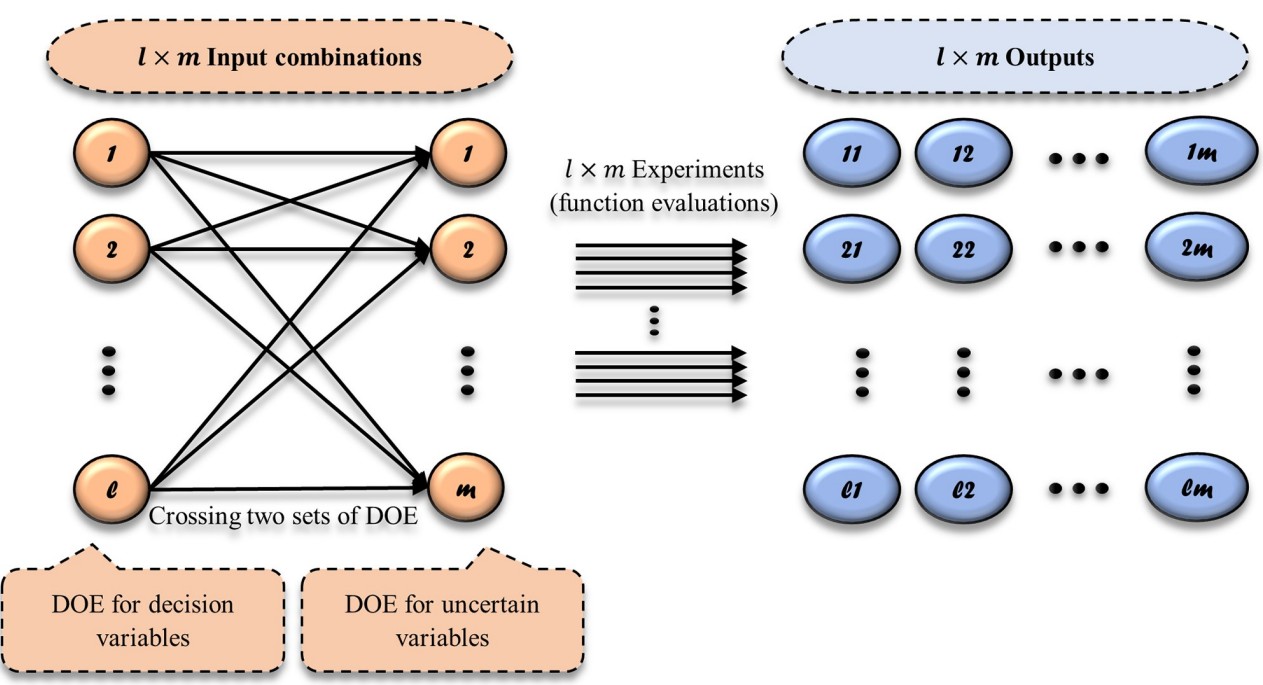

**Fig 4. Crossing two sets of DOE dealing with uncertainty in a model, one DOE (*l* samples) over decision variables of the model and second DOE (*m* samples) over uncertain variables in the model.**

obtained regarding Eq (6)), thus mean and standard deviation (Std) for each arrow in **Y** can be computed by the following equations:

$$Mean_s = \frac{1}{m}\sum_{r=1}^{m} y_{sr} \quad ,for \quad (s = 1, 2, \ldots, l) \tag{11}$$

$$Std_s = \sqrt{\frac{1}{m}\sum_{r=1}^{m} y_{sr}^2 - \left(\frac{1}{m}\sum_{r=1}^{m} y_{sr}\right)^2} \quad ,for \quad (s = 1, 2, \ldots, l) \tag{12}$$

Signal-to-Noise Ratio (SNR) as introduced by Taguchi [71, 72] is a robustness criterion based on the mean and the Std of a system response **Y**. Given that, **Y** is the smaller the better type, Taguchi assumed zero as the minimal possible response value. Accordingly, he formulated the following SNR as the robustness criterion:

$$SNR_s = 10\log[Mean_s 2 + \omega * Std_s 2] \quad ,for \quad (s = 1, 2, \ldots, l) \tag{13}$$

Since we performed a minimization of the model's output (here is the overall function, see Eq (6)) to find the optimal parameters of the FOPID controller, the formulation of the SNR in Eq (13) has the opposite sign by Taguchi formulation. Additionally, a weighting parameter $\omega$ is introduced to allow for individual emphasis on the minimization of variations. The smallest value of SNR in Eq (13) depicts the better point with smaller relevant simulation output and higher insensitivity to the source of variability (robustness).

**3.1.4 Efficient global optimization using a jackknife leave-one-out strategy.** A common formulation of efficient global optimization has been developed in the outline of the expected

improvement criterion, see [73, 74]. The expected improvement (EI) method has been developed in engineering design problems to adaptively improve the local and global search of optimal points (i.e. control a trade-off between exploration and exploitation properties) [75]. This method has been combined with two main parts. The first statistical part consists of the design of experiments and surrogate techniques and the second part involves evolutionary algorithms. If $SNR_c$ is considered for the arbitrary point $c$, an improvement function over the best point that is so far computed with $SNR_b$ is defined as $max\{0, (SNR_b − SNR_c)\}$. A common formulation of efficient global optimization in term of expected improvement creation is constructed as below:

$$EI(c) = (SNR_b − \widehat{SNR_c}) \, \Phi \left( \frac{SNR_b − \widehat{SNR_c}}{\widehat{S}_c} \right) + \widehat{S}_c \, \emptyset \left( \frac{SNR_b − \widehat{SNR_c}}{\widehat{S}_c} \right) \tag{14}$$

where $\widehat{S}_c$ indicates the estimation of GP perdition's error on candidate point $c$. The expression $SNR_b$ shows the value of the best signal to noise ratio that is obtained from true data of the original simulation model, and $\widehat{SNR_c}$ is GP surrogate prediction on candidate point $c$. The terms $\Phi$ and $\emptyset$ depict the cumulative distribution function (CDF) and probability density function (PDF) of a standard normal distribution respectively. The first phrase ($\Phi$) in Eq (14) is related to the local search and the second phrase ($\emptyset$) is related to a global search. In the search for the next best point among all the candidate points, the point with maximum EI term in Eq (14) is selected and replaced with the best point so far obtained. This procedure is continued until $Max \, REI − 0 \leq \varepsilon$, where $\varepsilon$ is a user-defined threshold, or an allocated computational cost (e.g. fixed number of repetitions) is reached. However, to find the neighbor points (candidate points) around the current best point, different strategies of sampling design methods can be used such as factorial designs [76] and space-filling design [77]. Here, we apply PSO global optimizer to investigate the maximum EI among the whole design space.

In order to estimate the surrogate prediction error for $c$th candidate point $(\widehat{S}_c)$ in Eq (14), simulation experiments can be resampled [73, 74]. The authors in [77] have used the bootstrapping technique to obtain perdition error for a GP surrogate using resampling to refit surrogate and obtain prediction error. However, resampling imposes extra computational cost due to the additional number of required simulation experiments (function evaluations). Here, to estimate the surrogate prediction error for each candidate point, we apply the jackknife leave-one-out approach. This approach applied an available set of I/O data and doesn't need resampling and extra simulation experiments.

**3.1.5 Jackknife leave-one-out approach.** Jackknife was first introduced by Quenouille (1949) [78] and named by Tukey (1958) [79]. The application of the jackknife method involves a leave-one-out strategy for the estimation of a parameter (e.g. the variance) in a dataset [80]. In this study, we are motivated to use the jackknife leave-one-out approach to estimate surrogate prediction error $(\widehat{S}_c)$ required in Eq (14) formulation, because this method uses an existing data and does not require to re-run the expensive simulation model. Here, this method is used to predict GP prediction error while it can be used for other surrogates as well. Let $\widehat{SNR_c}$ denotes the prediction of GP surrogate that fitted over all $l$ samples (input combinations), therefore the GP perdition error in $c$th candidate point $(\widehat{S}_c)$ can be estimated through the jackknife leave-one-out approach as the steps in Algorithm 1.

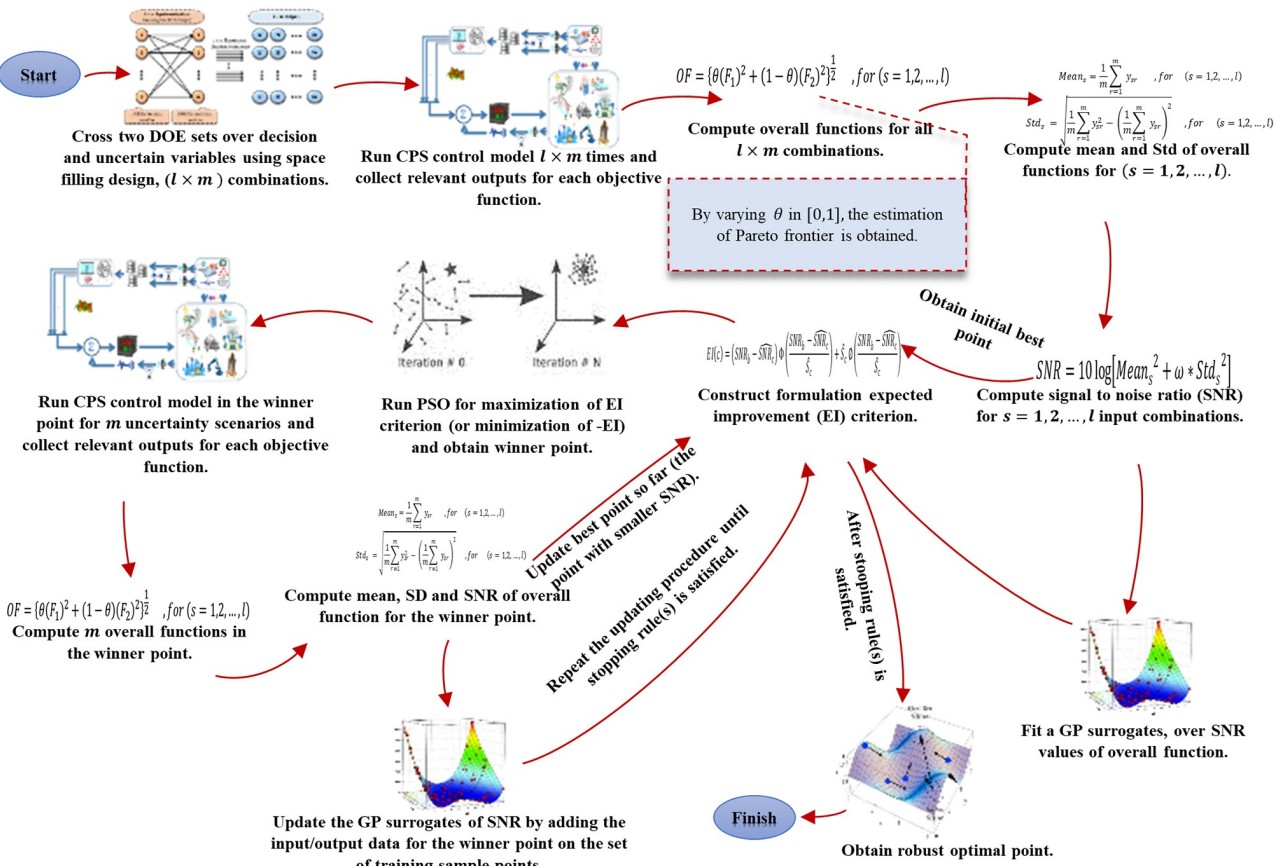

**Fig 5. Algorithmic representation of proposed approach for hybrid GP-PSO based robust simulation-optimization under uncertainty.**

## 3.2 Algorithmic framework

In this study, we develop a new hybrid surrogate/metaheuristic method applied in robust efficient global optimization and optimization under uncertainty. We apply a PSO metaheuristic to update a GP surrogate for sequential investigation of a robust optimal point. The proposed algorithm can handle robust efficient global optimization by the exhaustive search method that can be applied in real operation of CPS control frameworks. The algorithmic representation of the proposed approach is presented in Fig 5. The main steps involved in the proposed algorithm are presented in Algorithm 2. Note that, we assume the approximation function fitted over environmental factors $\widehat{f}(\tilde{z}_1(t), \tilde{z}_2(t), \ldots, \tilde{z}_n(t))$ can be used to estimate upper ($U_s$) and lower bound ($L_s$) for $\widehat{s}(t)$ by varying $\tilde{z}_1(t), \tilde{z}_2(t), \ldots, \tilde{z}_n(t)$ in their relevant ranges (upper and lower bounds of each relevant environmental factor) in time-step $t$. Here, these bounds are predefined and existed as inputs of the program.

Algorithm 1. Jackknife leave-one-out approach.

```
Input: Set of input combinations and relevant output (SNR).
Output: Estimation of surrogate prediction error for cth candidate
point.
begin
  Step 1. Select lc samples from the complete set of l combinations
(s = 1, 2, ...,l) when ic = l - k and k is a set of samples located in
vertices (i.e. we aim to avoid extrapolating of GP surrogate).
```

**Step2:** Drop **uth** samples (simulation experiment) and relevant SNR output when (**u = 1, 2, ..., $i_c$**).

**Step 3:** Fit a new GP surrogate over (**$l_c$ − 1 + k**) remaining samples.

**Step 4:** Predict output for **c**th candidate point ($\widehat{SNR}_c^{-u}$) using the GP surrogate constructed from the previous step.

**Step 5:** Implement three previous steps for all **$l_c$** samples computing **$l_c$** relevant predictions.

**Step 6:** Apply the jackknife estimator to obtain the estimation of surrogate prediction error for **c**th candidate point as below:

$$\widehat{S}_c \approx \left\{ \frac{1}{l_c} \sum_{u=1}^{l_c} (\widehat{SNR}_c - \widehat{SNR}_c^{-u})^2 \right\}^{1/2}$$

**End**

Algorithm 2. Proposed robust simulation-optimization approach.

**Input:** Estimated upper ($U_s$) and lower bound ($L_s$) for system's setpoint $\hat{s}(t) \in [L_s, U_s]$ and upper ($U_\alpha$) and lower bound ($L_\alpha$) for $\tilde{\alpha}$ due to anomaly in sensor feedback.

**Output:** The estimation of the Pareto frontier by a set of robust optimal points found by the proposed approach.

**begin**

**Step 1.** Design crossed array (using the space-filling design) by crossing two sets of experiments with dimensions **$l$ × $m$** as below:

- An inner array matrix with dimension ($l \times n_x$) where $l$ is the number of sample points for decision variables and $n_x$ is number of decision variables (e.g. in FOPID tuning $n_x = 5$ decision variables including three gain $K_i$, $K_p$, $K_d$ and two order λ, μ parameters).

- An outer array matrix with dimension (m × n_z) where m is the number of sample points (uncertainty scenarios) for n_z uncertain variables (e.g. here in represented CPS control system n_z = 2 including $\hat{s}(t)$ and $\tilde{\alpha}$).

**Step 2.** Run the CPS model (i.e. here we use simulation model) for each crossed (**$l$ × $m$**) combination and obtain the relevant output $\hat{y}_{sr}$ regarding each objective function, when **s = (1, 2, ..., $l$)** and **r = (1, 2, ..., $m$)**.

**Step 3.** Compute overall function (OF) values for all **$l$ × $m$** input combinations using Eq (6).

Step 4. Compute *Mean$_s$* and *Std$_s$* of overall function using Eqs (11) and (12) for each **s = (1, 2, ..., $l$)** sample point in inner array and compute relevant ***SNR$_s$*** using Eq (13).

**Step 5.** Fit a GP surrogate over sets of I/O data (with $l$ input combinations and relevant ***SNR$_s$*** values).

**Step 6.** Define an initial best point among the set of I/O data obtained from Step 4 (the point with the smallest SNR regarding Eq (13)).

**Step 7.** Set expected improvement criterion (see Eq (14)) as an objective function in PSO optimizer algorithm (i.e. with minimizing of −***EI (c)***) and obtaining a winner point.

**Step 8.** Run the real CPS model (e.g. original simulation model) in the winner point for **m** combinations of uncertainty (scenarios) designed in Step 1 and obtain relevant outputs for each objective function.

**Step 9.** Obtain OF values for the winner point regarding **m** uncertainty scenarios.

**Step 10.** Compute mean and Std of the winner point using Eqs (11) and (12).

**Step 11.** Update the set of I/O data **s = (1, 2, ..., $l$ + i)**, when **i** is the number of the sequential runs.

```
  Step 12. Fit a new GP surrogate over an updated set of I/O data (with
l + i training points and SNR as outputs).
  Step 13. Update (if needed) the best point obtained so far to a point
with smallest SNR ratio among all the sample points (including initial
training points and points which are added so far for updating of sur-
rogate, see Step 11) and repeat Step 7 till Step 12 until stopping
rules are satisfied (e.g. stop sequential updating if Max EI − 0 ≤ ε,
or i ≤ k, where ε and k are user-defined thresholds).
  Step 14. If stopping rule(s) is satisfied, then set the best point
obtained so far as a robust optimal point of the model. The best point
so far has the smallest SNR value among all sample points including
initial samples and updating sample points.
  Step 15. Obtain estimation of Pareto frontier by varying the weight
scale θ in [0,1] (see Eq (6)) and repeating Step 1 to Step 14.
end
```

In this study, we use a common space-filling design method named Latin hypercube sampling (LHS) with the desired correlational function to design simulation experiments. LHS was first introduced by McKay and colleagues [81]. It is a strategy to generate random sample points while guaranteeing that all the portions of the design space are depicted. LHS has been commonly defined for designing computer experiments based on the space-filling concept. In general, for $n$ input variables, $m$ sample points are produced randomly in $m$ intervals or scenarios (with equal probability). Inspired by [82] in the case of non-independent multivariate input variables, the desired correlation matrix can be used to produce distribution-free sample points in LHS. For more information, refer to [39, 83].

In the represented CPS control system in this study, outputs for two separate $F_1$ and $F_1$ objective functions need to be obtained regarding response error control and signal energy control, see Eqs (2) and (4). Notably, in Step 4, each $s = (1,2, \ldots, l)$ sample point is repeated $m$ times through $m$ different combinations (scenarios) of uncertain variables, see the framework of uncertainty management in Section 3.1.3. In Step 7, the fitted GP surrogate over SNR constructed in Step 5 is used to approximate the relevant SNR of each search point produced by PSO.

## 3.3 Augmented bootstrapping approach (sensitivity analysis)

In this study, the main idea behind the proposed algorithm is to perform sensitivity analysis to expand the information obtained from robust efficient global optimization. Estimating a single optimal point using a particular response may be inaccurate because of variability in the surrogate. Thus, we derive a series of possible responses that take into account a degree of uncertainty by providing confidence regions or prediction intervals. The author in [84] has mentioned two alternative strategies for bootstrapped resampling as follows:

- In each set of bootstrapping, both sets of input (design) combination ($X$) and noise (uncertain) combination ($Z$) are resampled randomly.

- The resampling is adapted to noise or uncertain component ($Z$) only while keeping the deterministic input combination ($X$) fixed.

Here, to find the bootstrapped set of data, a model is resampled $B$ times ($b = 1,2, \ldots, B$) (sampling with replacement), while $B$ is the number of resampled or bootstrapped sample size. Moreover, $B$ separate surrogates are fitted on $B$ different sets of sample points with the same size ($n$ design points). It is assumed that $d^+$ is a robust optimal solution which is obtained from the original (non-bootstrapped) surrogate. All output values in point $d^+$ are estimated using all the $B$ bootstrapped surrogates. The distribution-free bootstrapped Confidence Intervals (CIs)

can be computed as below, [59, 85]:

$$P(d^{+*}{}_{(\lfloor B(\gamma)\rfloor)} \leq d^+ \leq d^{+*}{}_{(\lceil B(1-(\gamma/2))\rceil)}) = 1 - \gamma \tag{15}$$

The superscript '*' is a common symbol for bootstrapped values [59]. The expression $\gamma$ gives two-sided CIs. Bonferroni's inequality suggests that Type I error rate for each interval per output is divided by the number of outputs (here is SNR). If the values of bootstrap estimate $SNR(d^+)^*$ are sorted from low to high, then $\lfloor.\rfloor$ and $\lceil.\rceil$ respectively denote floor and ceiling function to achieve the integer part and round upwards.

Here, inspired by [70, 86], the particular augmented bootstrapping approach is used for costly simulation running. In such a case, assume the set of sample points is fixed and only old data to fit surrogate with enough replication is available and new simulation replicating is very expensive. This augmented bootstrapping approach does not imply extra computational cost because of resampling and required simulation running to find a bootstrapped set of data. $x_s$ ($s = 1,2, \ldots, l$) denotes the set of sample points and each $x_s$ is repeated $m$ times ($r = 1,2, \ldots, m$). We assume that the original set of data obtained from the original simulation model is available (size $l \times m$) when $m$ is the number of scenarios for uncertainty and $l$ is the number of input combinations. Moreover, the augmented bootstrapping procedure is sketched in Algorithm 3.

Algorithm 3. The augmented bootstrapping procedure.

```
Input: Set of I/O data, and robust optimal point.
Output: Estimation of CIs.
begin
  Step 1. Set s = 1 and r = 1.
  Step 2. Choose (with replacement) one random number from the collec-
tion of {r* = 1, 2, ..., m}.
  Step 3. Replace the rth original output y_{s,r} (selected from the old
data) with the bootstrap outputy*_{s,r} = y_{s,r*}.
  Step 4. Set r = r + 1 and continue Step 3 and Step 4 till r = m.
  Step 5. Set s = s + 1 and continue Step 3, Step 4 and Step 5 till
s = l.
  Step 6: Compute Mean*_s, Std*_s, and SNR* using Eqs (11), (12) and (13)
respectively for (s = 1, 2, ...l) and fit a GP surrogate over new set of
I/O data.
  Step 7: Continue resampling B times (b = 1, 2, ..., B) where B is
the number of resampling or bootstrap sample size and compute
SNR*_b = (SNR*_1, SNR*_2, ..., SNR*_b).
  Step 8: Compute bootstrapped CIs using Eq (15) for the robust optimal
point obtained by the proposed algorithm as elucidated in Section
3.2).
end
```

Note that, regarding the Step 1 till Step 5, it can be seen that a random number with replacement in [1, $m$] is selected, and regarding the selected number, we choose the relevant response in an outer array (see the structure of the crossed array design explained in Section 3.1.3) that was previously collected from the original simulation model and has the same column number. For the same input combination, we repeat this procedure $m$ times and collect $m$ different responses or may have the same responses (i.e. because the random selection is done with replacement). This procedure is also repeated for other input combinations. Therefore, the data matrix with $l$ row and $m$ column is constructed.

## 4. Numerical example

Here, the proposed algorithm is specified for the robust optimal design of FOPID controller in CPS control of five-bar linkage robot manipulators. In the continue, we first explain the dynamics of the five-bar linkage robot manipulators. Next, the robust optimal design of FOPID controller in the CPS framework of a five-bar linkage robot manipulator is obtained using the proposed algorithm in this paper.

### 4.1 Dynamics of a five-bar linkage robot manipulator

Robotic manipulators, classic examples of nonlinear systems, are extensively used in the industry to automate various aspects of the production process of goods, thereby improving the quality of human life [87]. With the changing dynamics of these manipulators and their increasing complexity arising from their greater use, there has been considerable interest in their control technique fields. Robotic manipulators are Multi-Input Multi-Output (MIMO) systems with highly coupled nonlinear dynamics, posing a challenge to the development of their control scheme [88]. A five-bar linkage manipulator is a special class of parallel manipulators where a minimum of two kinematic chains control the motion of end-effectors [89]. The mechanism of a five-bar linkage is shown in Fig 6 [90].

Even though there are four links being moved, there are in fact only two degrees-of-freedom that are defined as $q_1$ and $q_2$. $q_i$ and $\tau_i$ are the joint variable and torque of the $i$th motor

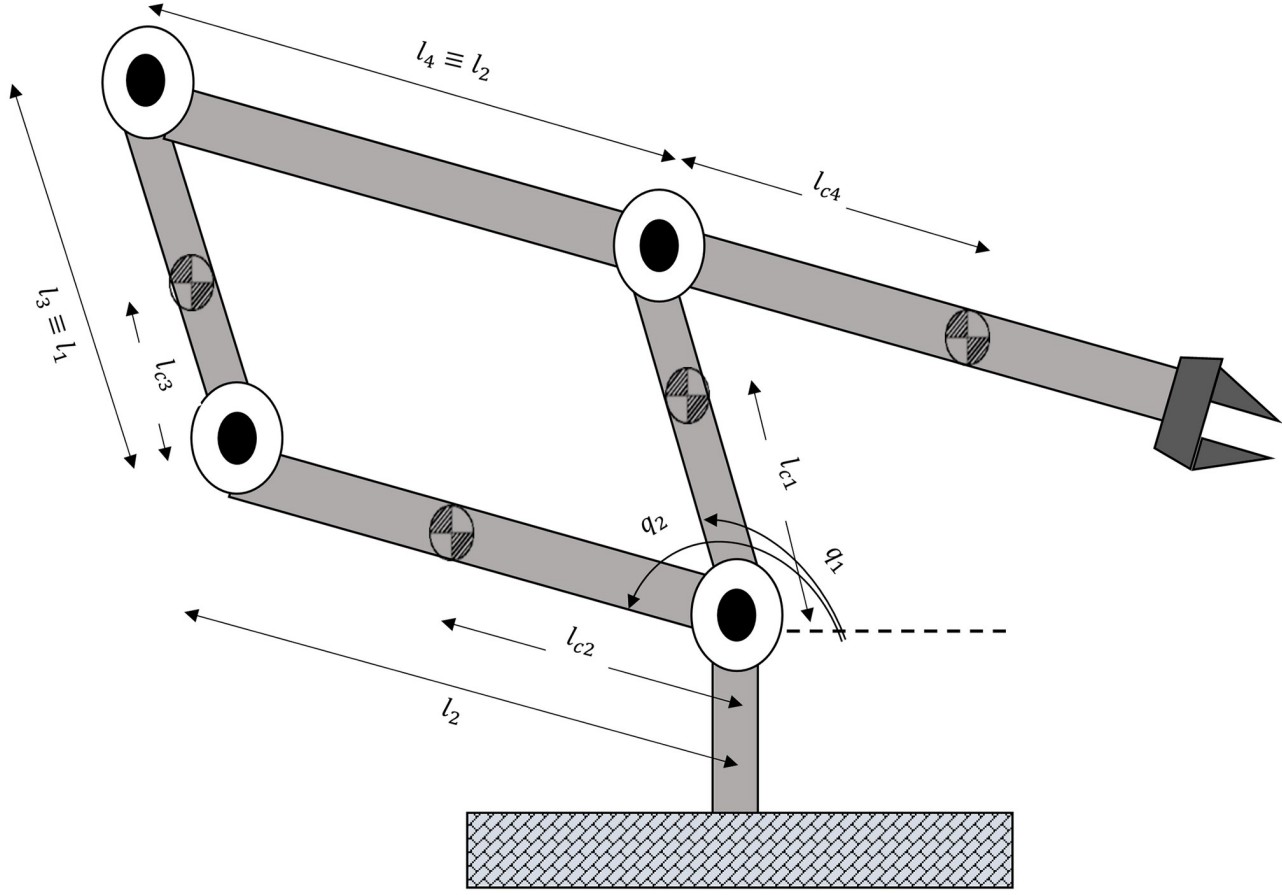

**Fig 6. Five bar linkage robot manipulator.**

respectively. Likewise, $I_i$, $l_i$, $l_{ci}$, and $m_i$ are the inertia matrix, length, distance to the center of gravity, and mass of the $i$th link respectively. In addition, if $m_3 l_2 l_{c3} = m_4 l_1 l_{c4}$, then the inertia matrix is diagonal and constant. As a consequence, the dynamic model of the manipulator is derived by the following equations [90]:

$$
\begin{aligned}
\tau_1 &= d_{11}\ddot{q}_1 + g\cos q_1(m_1 l_{c1} + m_3 l_{c3} + m_4 l_1) \\
\tau_2 &= d_{22}\ddot{q}_2 + g\cos q_2(m_1 l_{c1} + m_3 l_{c3} + m_4 l_1)
\end{aligned}
\tag{16}
$$

where g is gravitational constant and $d_{11}$ and $d_{22}$ are as follows:

$$
\begin{aligned}
d_{11} &= m_1 l_{c1}^2 + m_3 l_{c3}^2 + m_4 l_1^2 + I_1 + I_3 \\
d_{22} &= m_2 l_{c2}^2 + m_4 l_{c4}^2 + m_3 l_2^2 + I_2 + I_4
\end{aligned}
\tag{17}
$$

It should be noted that $\tau_1$ depends only on $q_1$ but not on $q_2$. Similarly, $\tau_2$ depends only on $q_2$ but not on $q_1$. This discussion helps to explain the popularity of the parallelogram configuration in industrial robots. If $m_3 l_2 l_{c3} = m_4 l_1 l_{c4}$, then two angles $q_1$ and $q_2$ can be adjusted independently without worrying about interactions between the two angles.

## 4.2 Simulation and algorithm setup

Here, the main goal is to obtain a robust optimal design of FOPID controller in such a CPS control model as elucidated in Section 2. We simulate the five-bar linkage robot manipulator using Eq (16) in Matlab®/Simulink environment. Simulink does not have a library for the FOPID. Therefore, the controller from the library of FOMCON: a Matlab® toolbox for fractional-order system identification and control [91], which allows for the computation of the fractional-order derivative and integration is used. Numeric values of the parameters of the five-bar manipulator dynamics are taken from [54, 92] as shown in Table 2. From the data that are driven in Table 2, it is revealed that $m_3 l_2 l_{c3} = m_4 l_1 l_{c4}$, thus we can perform robust optimal design of controller for one motor, where the same results are also valid for the second motor. In FOPID tuning, five decision variables including $K_i$, $K_p$, $K_d$, $\lambda$, and $\mu$ are considered as decision variables. The search procedure for the robust optimal result is done in the ranges as [54]:

$$
K_p \in [0, 30], K_i \in [0, 5], K_d \in [0, 5], \mu \in [0, 1] \text{and } \lambda \in [0, 1]
$$

Two performance criteria are considered as outputs of the model including Eqs (2) and (4) in time-step $t$ (here the size of time-step is fixed at 0.01) and time domain (simulation time) $T = 20$. In addition, for uncertain variables, we assume that $\widehat{s}(t)$ varies in [0.5,2.5] and $\tilde{\alpha}$ varies in [−0.05,0.05]. However, we implement the proposed algorithm in CPS control framework of a five-bar linkage robot manipulator.

The following process is done to determine the robust optimal values of the FOPID parameters ($K_i$, $K_p$, $K_d$, $\lambda$, and $\mu$) using the proposed algorithm. First, we design a set of experiments with the size of $l = 15$ samples using LHS. Another sampling design is constructed for

Table 2. Numeric values of the parameters of the five-bar manipulator dynamics.

| Link | Mass ($Kg$) | Length ($m$) | C of g ($m$) | Inertia ($Kgm^2$) |
|------|------------|-------------|-------------|-------------------|
| 1 | 0.2880 | 0.33 | 0.166 | 1 |
| 2 | 0.0324 | 0.12 | 0.060 | 2 |
| 3 | 0.3702 | 0.33 | 0.166 | 1 |
| 4 | 0.2981 | 0.45 | 0.075 | 2 |

uncertain factors $\widehat{s}(t)$ and $\tilde{\alpha}$ (here we choose $m = 9$ samples as the size of uncertainty scenarios). Two Matlab$^{\circledR}$ functions "lhsdesign" and "gridsamp" are used to design training sample points with minimum correlation and to design uncertainty scenarios (different combinations of uncertain factors) respectively. We cross both sets of experiments to follow the crossed array design framework as elaborated in Section 3.1.3. Each input combination in the inner array $s = (1,2, \ldots, l = 15)$ including designed values of $K_i$, $K_p$, $K_d$, $\lambda$, and $\mu$ are sent to Simulink block for $m$ times regarding each uncertainty scenarios $r = (1, 2, \ldots, m = 9)$ and the values of SEC and REC in time domain are collected. So, $15 \times 9$ simulation outputs are collected according to 135 simulation runs (function evaluations). We, use the collected data to obtain $F_1$, $F_2$, and, OF regarding Eqs (2), (4) and (6) respectively. We set both $M_1$ and $M_2$ parameters equal to 10 used in Eqs (2) and (4). Regarding $m$ uncertainty scenarios, we repeat each input combination $m$ times, and compute the relevant mean and Std of OF using Eqs (11) and (12). Then, we calculate $SNR_s$ for each input combination $s = (1,2, \ldots, l = 15)$ using Eq (13) while assume $\omega = 3$. Afterwards, we fit a GP surrogate over set of input combinations and set of $SNR_s$ outputs. The DACE [93], Matlab$^{\circledR}$ toolbox has been employed to construct GP surrogate. In the current study, first-order polynomial regression and Gaussian correlation functions are adjusted to fit GP surrogate. The correlation parameter is fixed on 0.1 (i.e. in the DACE toolbox, the correlation parameter is forced to vary in the range between 0.01 to 20).

Next, we perform the procedure of sequential expected improvement to estimate the robust optimal point after $n$ sequential EI iterations. Among all the $SNR$ values in the set of $SNR_s$, a sample with the smallest $SNR$ value and its relevant input combination including the relevant vector of $[K_i, K_p, K_d, \lambda, \mu]$ is considered as an initial best point. Regarding our proposed algorithm, we apply the PSO optimizer to search for a winner point in each sequential EI iteration. For setting the PSO parameters, the maximum iteration number is fixed 200 and the swarm is initialized with 30 particles. Notably, as we use GP surrogate instead of the true (original) simulation model as an objective function in PSO, thus we don't worry about the computational cost due to running of a true simulation model. At the end of any sequential EI iteration, the program checks the stopping rule(s). Here, we stop the EI procedure when the EI criterion becomes smaller than 0.01, or the number of sequential runs reaches 15 iterations. Also, at the end of any sequential EI iteration, two terms of the program are updated, i) the set of training sample points by adding a winner point and relevant SNR output that is computed accordingly from the original simulation model, ii) the best sample point obtained so far with the smallest SNR among all the training points and updating points. Moreover, according to an updated set of training samples, a new GP surrogate is constructed after each sequential EI iteration. It is important to note that we avoid extrapolation of GP surrogate in each sequential iteration by setting two different rules, i) we consider a death penalty for any point that is investigated by PSO and is located out of bounds of training points, ii) to estimate GP prediction error using jackknife leave-one-out approach, we only remove input combination's rows that don't locate on the margin of design space (see Section 3.1.5). The obtained results from the proposed algorithm and relevant sensitivity analysis are discussed in the following sections.

## 4.3 Robust optimal results

We perform the proposed algorithm for three different values of $\theta = 0.25$, $\theta = 0.5$, and $\theta = 0.75$ in computing OF (see Eq (6)). To evaluate the effect of randomness in sampling design methods, each optimization set was repeated 10 times. Tables 3–5 show the obtained results using the proposed algorithm for estimating robust FOPID optimal design over 10 different repetitions for $\theta = 0.25$, $\theta = 0.5$, and $\theta = 0.75$ respectively. As mentioned in Section 4.2, the obtained SNR values are computed by repeating each set of FOPID gain parameters over 9 different

**Table 3. Robust FOPID optimal results using proposed algorithm for 10 repetitions for θ = 0.25 (the results obtained over 9 different uncertainty scenarios).**

| Repeat No | FOPID Optimal Parameters | | | | | Simulation Experiments | | | Optimum SNR value | | |
|---|---|---|---|---|---|---|---|---|---|---|---|
| | $K_p$ | $K_i$ | $K_d$ | $\mu$ | $\lambda$ | Total | In.sa | Up.sa | SNR ($\omega$ = 3) | Ave | Std |
| 1 | 4.133 | 1.411 | 1.839 | 0.967 | 0.967 | 171 | 135 | 36 | -20.564 | -20.263 | 0.219 |
| 2 | 1.000 | 2.638 | 2.925 | 0.033 | 0.790 | 180 | 135 | 45 | -20.209 | | |
| 3 | 3.321 | 1.911 | 3.875 | 0.033 | 0.804 | 198 | 135 | 63 | -19.911 | | |
| 4 | 1.000 | 4.494 | 2.854 | 0.875 | 0.831 | 171 | 135 | 36 | -20.143 | | |
| **5** | **1.000** | **2.250** | **2.559** | **0.387** | **0.852** | **207** | **135** | **72** | **-20.621** | | |
| 6 | 3.311 | 2.387 | 3.206 | 0.305 | 0.896 | 207 | 135 | 72 | -20.119 | | |
| 7 | 1.509 | 0.167 | 2.922 | 0.458 | 0.615 | 216 | 135 | 81 | -20.208 | | |
| 8 | 1.000 | 2.182 | 3.430 | 0.575 | 0.726 | 198 | 135 | 63 | -20.527 | | |
| 9 | 3.311 | 2.387 | 3.206 | 0.305 | 0.896 | 207 | 135 | 72 | -20.119 | | |
| 10 | 1.509 | 0.167 | 2.922 | 0.458 | 0.615 | 216 | 135 | 81 | -20.208 | | |

**Table 4. Robust FOPID optimal results using proposed algorithm for 10 repetitions for θ = 0.5 (the results obtained over 9 different uncertainty scenarios).**

| Repeat No | FOPID Optimal Parameters | | | | | Simulation Experiments | | | Optimum SNR value | | |
|---|---|---|---|---|---|---|---|---|---|---|---|
| | $K_p$ | $K_i$ | $K_d$ | $\mu$ | $\lambda$ | Total | In.sa | Up.sa | SNR ($\omega$ = 3) | Ave | Std |
| 1 | 3.321 | 1.549 | 2.634 | 0.967 | 0.967 | 225 | 135 | 90 | -21.838 | -21.840 | 0.245 |
| **2** | **1.000** | **0.713** | **2.017** | **0.802** | **0.726** | **261** | **135** | **126** | **-22.249** | | |
| 3 | 1.000 | 2.668 | 3.604 | 0.575 | 0.832 | 225 | 135 | 90 | -21.879 | | |
| 4 | 1.000 | 1.228 | 2.436 | 0.657 | 0.844 | 189 | 135 | 54 | -22.120 | | |
| 5 | 1.000 | 2.667 | 3.947 | 0.696 | 0.757 | 225 | 135 | 90 | -21.949 | | |
| 6 | 3.429 | 1.782 | 3.410 | 0.298 | 0.861 | 243 | 135 | 108 | -21.562 | | |
| 7 | 7.000 | 1.833 | 2.833 | 0.433 | 0.967 | 198 | 135 | 63 | -21.448 | | |
| 8 | 1.000 | 1.964 | 3.909 | 0.775 | 0.705 | 180 | 135 | 45 | -22.003 | | |
| 9 | 3.740 | 1.702 | 2.151 | 0.033 | 0.915 | 189 | 135 | 54 | -21.551 | | |
| 10 | 2.233 | 1.624 | 2.000 | 0.318 | 0.967 | 180 | 135 | 45 | -21.803 | | |

uncertainty scenarios designed by the grid sampling method. In those tables, two expressions "In.sa" and "Up.sa" indicate the initial sampling design and updating samples that are added to the training set through the procedure of sequential improvement respectively.

As can be seen, for θ = 0.25, the best SNR (−20.621) is obtained through the fifth repetition with a total of 207 function evaluations (15 × 9 runs regarding initial crossed sampling design

**Table 5. Robust FOPID optimal results using proposed algorithm for 10 repetitions for θ = 0.75 (the results obtained over 9 different uncertainty scenarios).**

| Repeat No | FOPID Optimal Parameters | | | | | Simulation Experiments | | | Optimum SNR value | | |
|---|---|---|---|---|---|---|---|---|---|---|---|
| | $K_p$ | $K_i$ | $K_d$ | $\mu$ | $\lambda$ | Total | In.sa | Up.sa | SNR ($\omega$ = 3) | Ave | Std |
| 1 | 8.238 | 2.300 | 3.748 | 0.426 | 0.919 | 261 | 135 | 126 | -23.868 | -24.041 | 0.140 |
| **2** | **1.267** | **3.101** | **3.332** | **0.525** | **0.835** | **270** | **135** | **135** | **-24.238** | | |
| 3 | 4.285 | 1.998 | 3.510 | 0.612 | 0.938 | 189 | 135 | 54 | -24.032 | | |
| 4 | 7.803 | 3.239 | 3.257 | 0.944 | 0.918 | 216 | 135 | 81 | -24.191 | | |
| 5 | 1.025 | 2.144 | 3.072 | 0.567 | 0.860 | 270 | 135 | 135 | -24.151 | | |
| 6 | 6.113 | 1.314 | 3.039 | 0.709 | 0.967 | 243 | 135 | 108 | -24.022 | | |
| 7 | 5.775 | 2.308 | 3.689 | 0.348 | 0.913 | 234 | 135 | 99 | -23.879 | | |
| 8 | 5.201 | 2.485 | 4.124 | 0.823 | 0.794 | 261 | 135 | 126 | -24.213 | | |
| 9 | 5.775 | 2.308 | 3.689 | 0.348 | 0.913 | 234 | 135 | 99 | -23.879 | | |
| 10 | 11.00 | 1.833 | 2.833 | 0.900 | 0.967 | 270 | 135 | 135 | -23.939 | | |

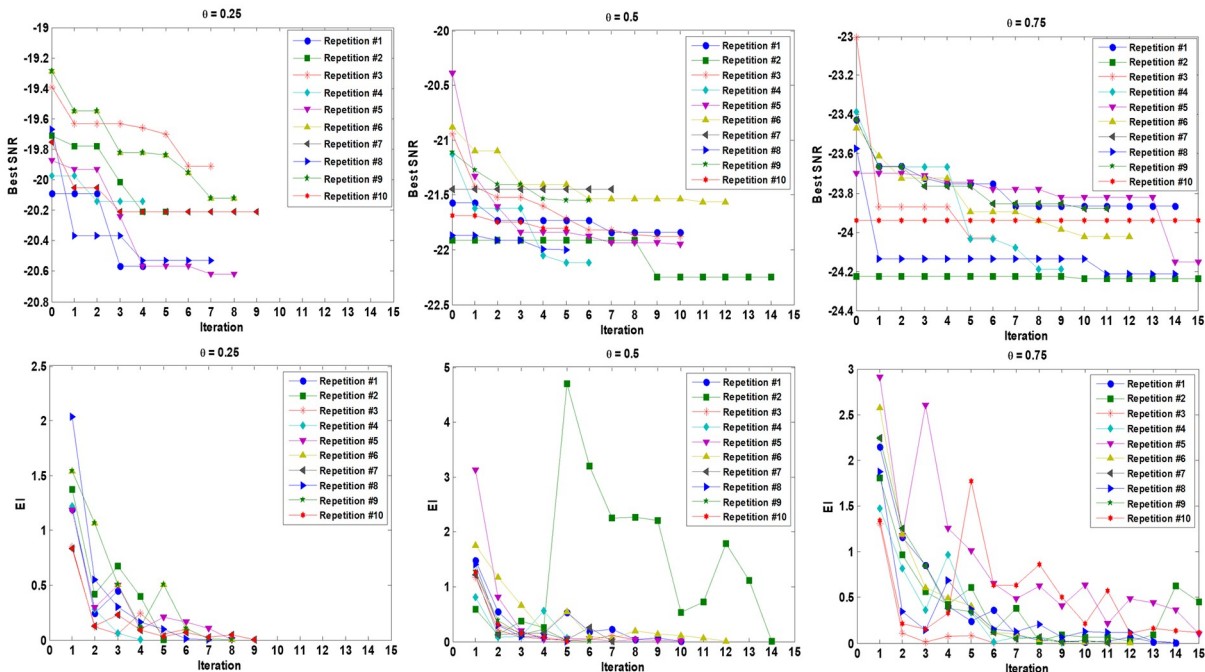

**Fig 7. EI criterion magnitudes and best SNR obtained by sequential expected improvement over 10 different repetition of proposed algorithm for $\theta = 0.25$, $\theta = 0.5$ and $\theta = 0.75$.** Two stopping rules are adjusted, EI value becomes smaller than 0.01 or reach 15 sequential iterations.

and $8 \times 9$ simulation runs regarding sequential updating of the training sample set). For $\theta = 0.5$, the best SNR (-22.249) is obtained from the second repetition and with a total of 261 simulation experiments (135 initial samples plus 126 updating samples). The best SNR value (-24.238) for $\theta = 0.75$ is obtained through the second repetition by a total of 270 function evaluations (15 initial input combinations and 15 update combinations that are crossed by 9 uncertainty scenarios). We consider all the three best points over all 10 repetitions as robust optimal points for the FOPID controller using the proposed algorithm for each $\theta = 0.25$, $\theta = 0.5$, and $\theta = 0.75$ separately. Fig 7 shows the magnitudes of the EI criterion and the best SNR obtained by sequential expected improvement over 10 different repetitions of the proposed algorithm for $\theta = 0.25$, $\theta = 0.5$, and $\theta = 0.75$. Also, the mean and Std of OF related to the best point so far (smaller SNR) in each sequential EI iteration are shown in Fig 8. It should be noted that two stopping rules are adjusted, the EI value becomes smaller than 0.01, or the sequential procedure reaches 15 sequential iterations. Fig 9 shows the step responses of the robot manipulator with 9 different uncertainty scenarios ($\widehat{s}(t) = [0.5, 1.5, 2.5]$ and $\tilde{\alpha} = [-0.05, 0, +0.05]$) for $\theta = 0.25$, $\theta = 0.5$, and $\theta = 0.75$.

## 4.4 Sensitivity analysis

To analyze the sensitivity of robust optimal results obtained by the proposed algorithm and estimate the variability which occurred due to randomness in designing sample points, we used the augmented bootstrapping method explained in Section 3.3. Here, based on the obtained results from the original GP surrogate, the FOPID parameters in robust optimum point ($d^+$) for $\theta = 0.25$, $\theta = 0.5$, and $\theta = 0.75$ are defined as below:

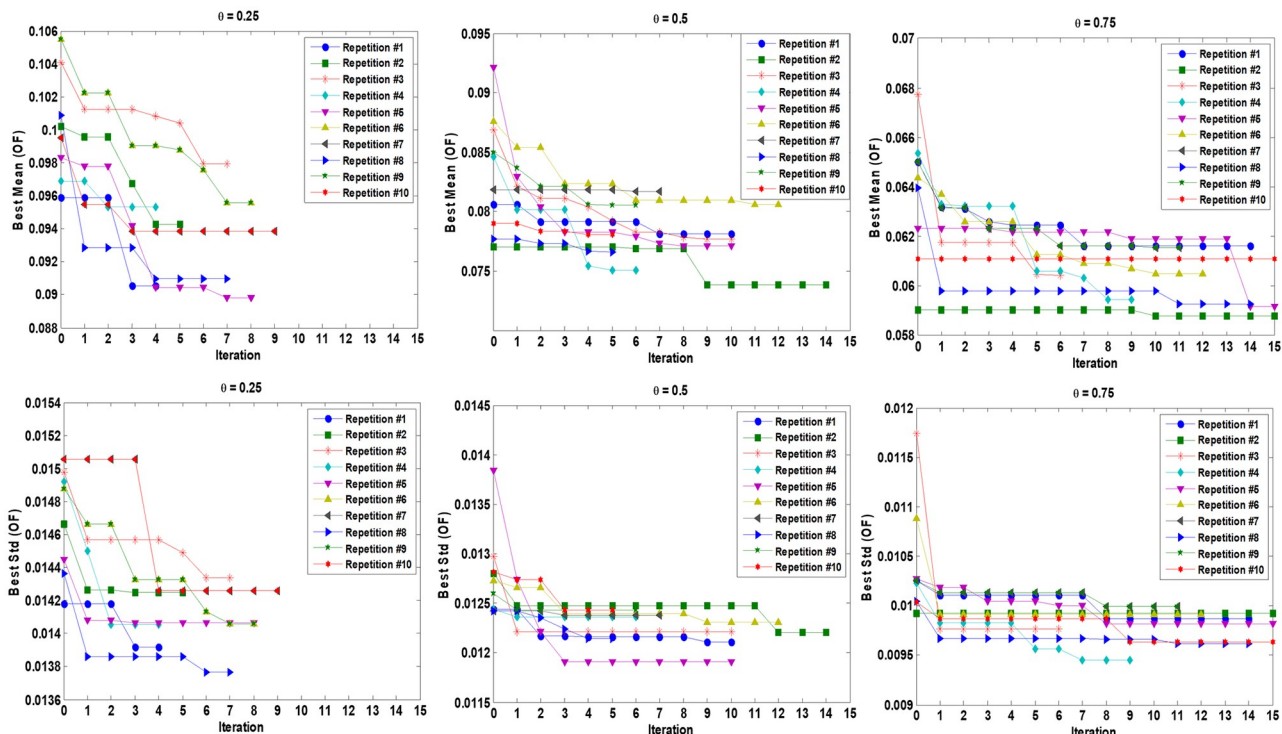

**Fig 8. Mean and Std of overall function (OF) related to best point so far (smaller SNR) obtained by sequential expected improvement over 10 different repetition of proposed algorithm for $\theta = 0.25$, $\theta = 0.5$ and $\theta = 0.75$.** Two stopping rules are adjusted, EI value becomes smaller than 0.01 or reach 15 sequential iterations.

- For, $\theta = 0.25$

$$d^+ = \{K_p = 1.00, K_i = 2.250, K_d = 2.259, \mu = 0.387 \text{ and } \lambda = 0.852\}$$

$$SNR(d^+) = -20.621$$

- For $\theta = 0.5$

$$d^+ = \{K_p = 1.00, K_i = 0.713, K_d = 2.017, \mu = 0.802 \text{ and } \lambda = 0.726\}$$

$$SNR(d^+) = -22.249$$

- For $\theta = 0.75$

$$d^+ = \{K_p = 1.267, K_i = 3.101, K_d = 3.332, \mu = 0.525 \text{ and } \lambda = 0.835\}$$

$$SNR(d^+) = -24.238$$

With $B$ predicted values from $B$ bootstrapped GP surrogates, we can quantify the CIs (bootstrapped confidence intervals) for $d^+$. For the current case, we selected the bootstrapped size $B = 50$. We predicted SNR by each $B = 50$ bootstrapped surrogates in the robust optimal point which are obtained by original GP surrogates. From these 50 values for SNR, we estimated CIs for SNR by applying Eq (15). We quantified these confidence regions for $\gamma = 0.05$ (i.e. $\gamma$

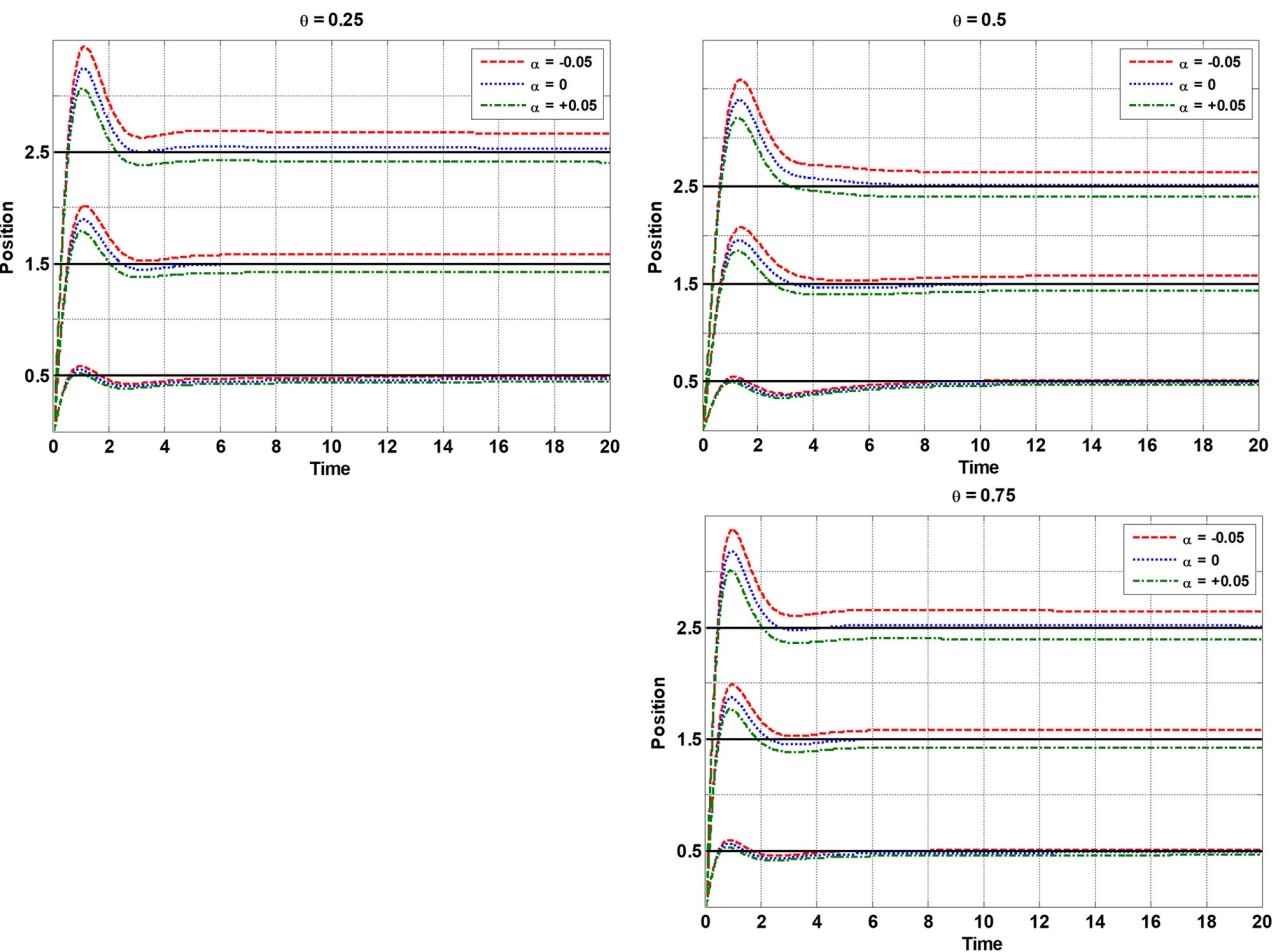

**Fig 9. The step responses of the robot manipulator with 9 different uncertainty scenarios ($\widehat{s}(t) = [0.5, 1.5, 2.5]$ and $\tilde{\alpha} = [-0.05, 0, +0.05]$) for $\theta = 0.25$, $\theta = 0.5$ and $\theta = 0.75$.**

denotes type I error and shows the probability of becoming infeasible from estimated confidence regions). As we estimated the robustness as a consequence of the uncertainty in the model, it becomes important to implement further analyses of the statistical variation. The CIs shows that the original estimated SNR may still give variety regarding its threshold due to the variability of surrogates' predictions. However, 95% two-sided approximations of CIs obtained by bootstrapped GP surrogates for SNR values in robust optimal points of FOPID controller regarding $\theta = 0.25$, $\theta = 0.5$, and $\theta = 0.75$ are as follows:

$$P\left(E(d^+)^*_{(\lfloor 50(0.05/2)\rfloor)} \leq E(d^+) \leq E(d^+)^*_{(\lceil 50(1-(0.05/2))\rceil)}\right) = 0.95$$

- For $\theta = 0.25$
  Lower bound: $E(d^+)^*_{(\lfloor 50(0.05/2)\rfloor)} = -21.566$
  Upper bound: $E(d^+)^*_{(\lceil 50(1-(0.05/2))\rceil)} = -20.060$

- For $\theta = 0.5$
  Lower bound: $E(d^+)^*_{(\lfloor 50(0.05/2)\rfloor)} = -23.431$
  Upper bound: $E(d^+)^*_{(\lceil 50(1-(0.05/2))\rceil)} = -21.585$

- For $\theta = 0.75$
  Lower bound: $E(d^+)^*_{(\lfloor 50(0.05/2)\rfloor)} = -25.152$
  Upper bound: $E(d^+)^*_{(\lceil 50(1-(0.05/2))\rceil)} = -23.517$

Figs 10–12 show the sensitivity analysis obtained by bootstrapping for $\theta = 0.25$, $\theta = 0.5$, and $\theta = 0.75$ respectively regarding FOPID gain parameters ($K_p$, $K_i$, $K_d$) and order parameters ($\mu$, $\lambda$).

## 4.5 Proposed algorithm versus three common optimizers in FOPID tuning

In this section, we compare the results obtained by a proposed algorithm with three common FOPID optimization methods including the PSO metaheuristic [62] (i.e. when directly used in optimization procedure), Grey Wolf Optimizer (GWO) [94], and Ant Lion Optimizer (ALO) [95]. These methods have been widely used in the literature for optimal control systems [49, 54, 96, 97]. We compare both levels of accuracy (lower objective function) and the robustness of each method in the tuning of stochastic controllers. Here, we assume that the model is limited to only 270 simulation experiments (function evaluations) to obtain a robust optimal design of stochastic FOPID controller for $\theta = 0.25$, $\theta = 0.5$, and $\theta = 0.75$. So, we let each optimizer employ a maximum of 270 simulation experiments. It should be noted that we also allowed our proposed algorithm to use maximum 270 simulation experiments to search for the optimal point, see Section 4.2. The parameters settings for all the three optimizers (PSO, GWO, and ALO) are as follows:

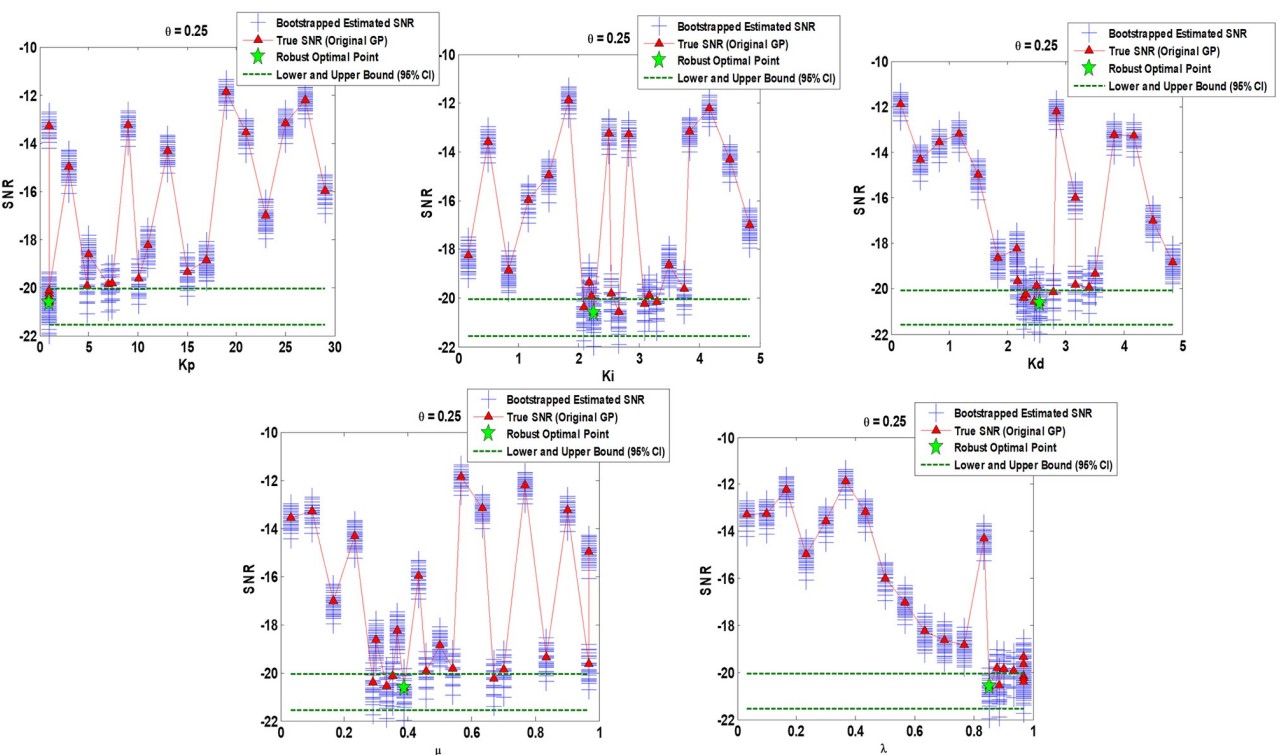

**Fig 10. Sensitivity analysis via 50 bootstrapped GP surrogate and 95% Confidence Intervals (CIs) over robust optimal point obtained by original GP surrogate for $\theta = 0.25$.** Augmented parametric bootstrapping is performed using on hand set of input/output data provided among original optimization program.

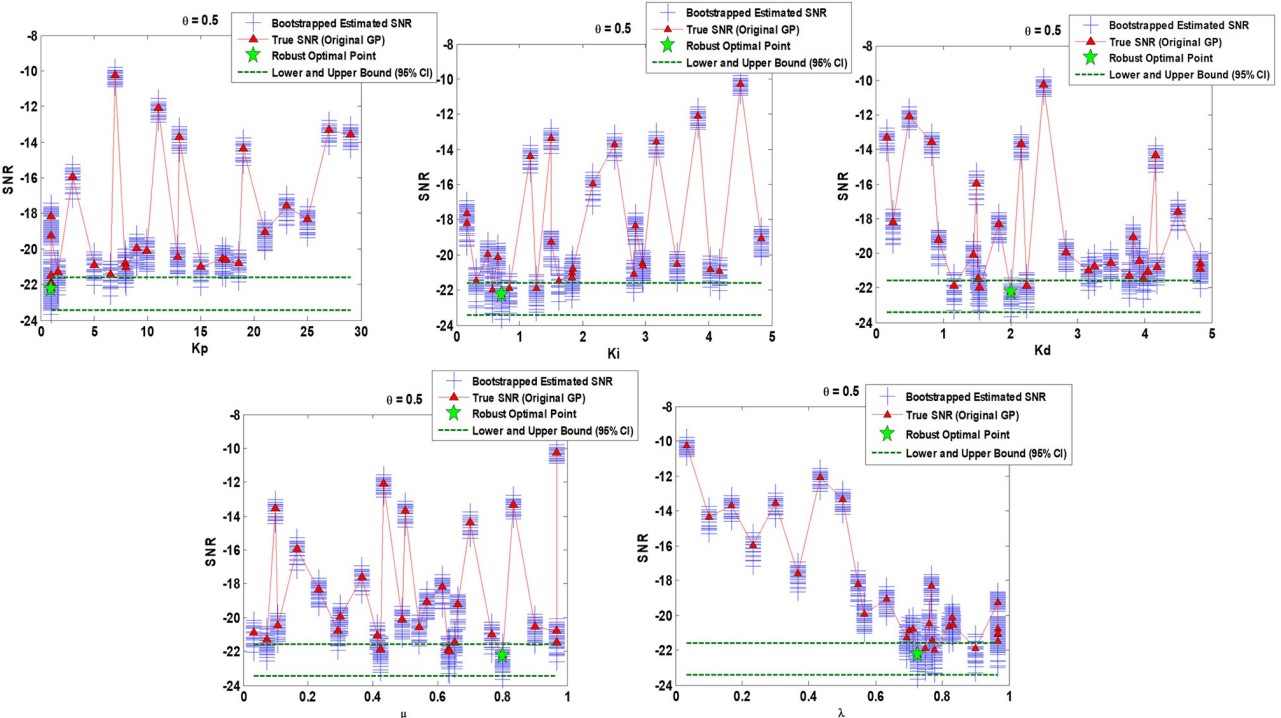

**Fig 11. Sensitivity analysis via 50 bootstrapped GP surrogate and 95% Confidence Intervals (CIs) over robust optimal point obtained by original GP surrogate for θ = 0.5.** Augmented parametric bootstrapping is performed using on hand set of input/output data provided among original optimization program.

The number of iterations is considered to be 30 and initial papulation is adjusted to 9. The other parameters for PSO [62, 63] are also selected as follows:

Min of inertia weight equals to 0.4; max of inertia weight equals to 0.9; all the three factors of velocity clamping factor, cognitive constant, and social constant are set to 2. To run each of the three mentioned optimizers in each relevant iteration by optimizer, we randomly (with replacement) produce a scenario of uncertainty and compute output of the original simulation including SEC and REC and compute OF as an objective (fitness) function of optimizer. To make a fair comparison between the proposed algorithm and three optimizers (PSO, GWO, and ALO) in the stochastic FOPID control system, we repeat each of the three optimizers 10 times (as mentioned in Section 4.2 and 4.3, we repeated the proposed algorithm 10 times). To compare the obtained results using proposed algorithm and the three global optimizers (PSO, GWO, and ALO), we produce 100 different combinations (scenarios) of two uncertain factors including $\hat{s}(t)$ and $\tilde{\alpha}$ using grid sampling design approach. Afterwards, for each set of optimal FOPID parameters according to the obtained results by proposed algorithm and three global optimizers, we run true simulation model regarding each uncertainty scenario (total 100 simulation runs for each set of FOPID optimal point).

**4.5.1 Comparison results.** Tables 6–8 provide the statistical comparison results between the proposed algorithm and three common optimizers in 10 separate repetitions for **θ = 0.25, θ = 0.5, and θ = 0.75** respectively. In these tables, the level of accuracy (lower objective function) and robustness for the stochastic FOPID tuning are compared. The FOPID tuning results using different methods are obtained over 100 different uncertainty scenarios. Note that the expression "SE" in the tables indicates the total number of simulation experiments (function evaluations) employed for the optimization procedure. It should be also noted that we allow

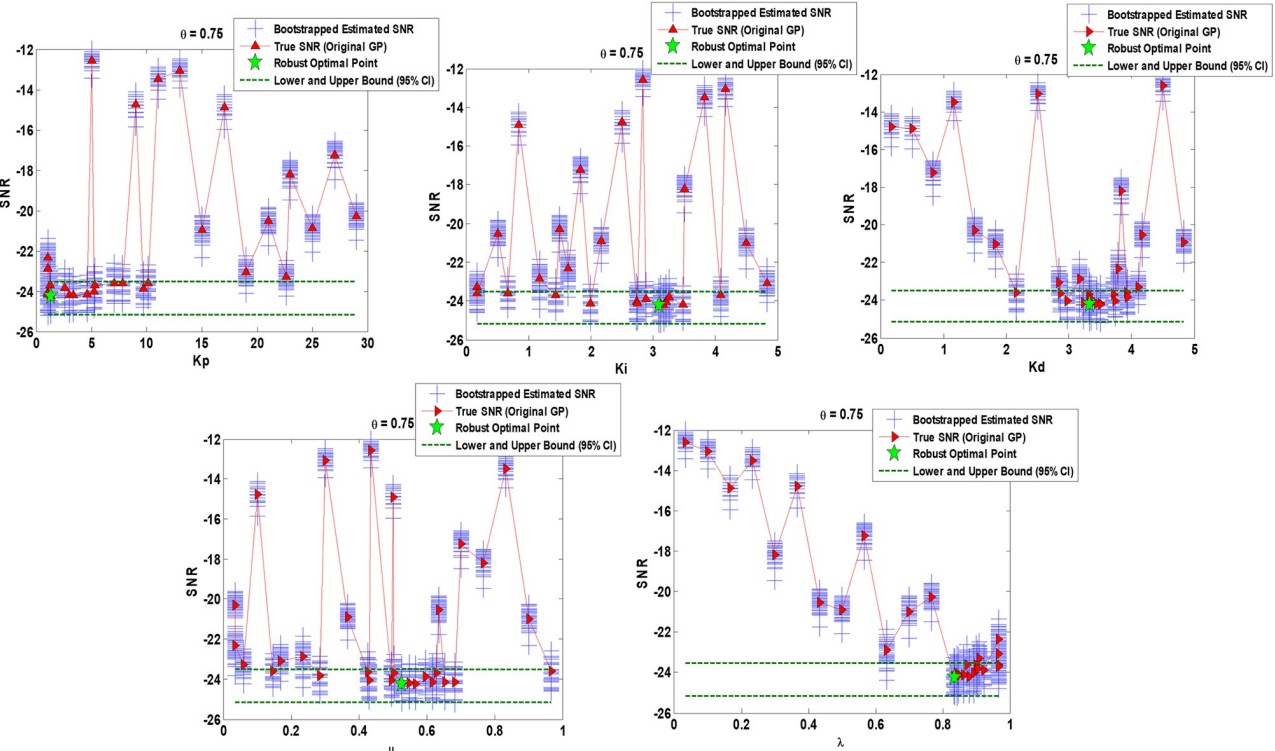

**Fig 12. Sensitivity analysis via 50 bootstrapped GP surrogate and 95% Confidence Intervals (CIs) over robust optimal point obtained by original GP surrogate for θ = 0.75.** Augmented parametric bootstrapping is performed using on hand set of input/output data provided among original optimization program.

each optimizer to use a maximum of 270 function evaluations. For the proposed algorithm, we consider two stopping rules for the relevant procedure of sequential improvement. The EI criterion becomes smaller than 0.01 or the algorithm reaches 15 sequential iterations (See Section 4.2).

As can be seen from Table 6, the lower SNR is obtained by GWO with -20.944, and it is followed by our proposed algorithm with -20.499. The PSO and ALO optimizers also provided competitive results. Regarding the effect of randomness on 10 repetitions (stochastic

**Table 6. Comparison results for FOPID tuning using different methods over 100 different uncertainty scenarios for θ = 0.25.**

| Repeat No | Proposed algorithm | | | PSO | | | GWO | | | ALO | | |
|---|---|---|---|---|---|---|---|---|---|---|---|---|
| | SNR | Ave / Std | SE | SNR | Ave / Std | SE | SNR | Ave / Std | SE | SNR | Ave / Std | SE |
| 1 | -20.806 | -20.499 / 0.238 | 1971 | -19.325 | -20.404 / 0.630 | 2700 | -21.351 | -20.944 / 0.545 | 2700 | -20.666 | -20.205 / 0.642 | 2700 |
| 2 | -20.450 | | | -20.620 | | | -21.338 | | | -18.721 | | |
| 3 | -20.112 | | | -20.337 | | | -21.076 | | | -20.131 | | |
| 4 | -20.349 | | | -19.647 | | | -20.877 | | | -20.066 | | |
| 5 | -20.894 | | | -20.352 | | | -19.477 | | | -20.742 | | |
| 6 | -20.329 | | | -21.162 | | | -21.484 | | | -20.736 | | |
| 7 | -20.470 | | | -20.841 | | | -20.695 | | | -20.217 | | |
| 8 | -20.782 | | | -19.806 | | | -21.036 | | | -20.460 | | |
| 9 | -20.329 | | | -21.426 | | | -20.841 | | | -20.875 | | |
| 10 | -20.470 | | | -20.524 | | | -21.260 | | | -19.434 | | |

**Table 7. Comparison results for FOPID tuning using different methods over 100 different uncertainty scenarios for θ = 0.5.**

| Repeat No | Proposed algorithm | | | PSO | | | GWO | | | ALO | | |
|---|---|---|---|---|---|---|---|---|---|---|---|---|
| | SNR | Ave / Std | SE | SNR | Ave / Std | SE | SNR | Ave / Std | SE | SNR | Ave | SE |
| 1 | -22.076 | -22.112 / 0.280 | 2115 | -20.915 | -21.101 / 1.066 | 2700 | -22.376 | -22.165 / 0.377 | 2700 | -21.801 | -21.824 / 0.562 | 2700 |
| 2 | -22.616 | | | -21.053 | | | -22.470 | | | -21.903 | | |
| 3 | -22.136 | | | -19.600 | | | -21.797 | | | -21.780 | | |
| 4 | -22.448 | | | -19.120 | | | -22.417 | | | -22.017 | | |
| 5 | -22.201 | | | -20.193 | | | -21.358 | | | -21.806 | | |
| 6 | -21.810 | | | -22.173 | | | -22.372 | | | -21.824 | | |
| 7 | -21.653 | | | -22.005 | | | -22.550 | | | -22.540 | | |
| 8 | -22.262 | | | -21.965 | | | -22.257 | | | -20.281 | | |
| 9 | -21.821 | | | -22.057 | | | -22.338 | | | -22.216 | | |
| 10 | -22.096 | | | -21.929 | | | -21.714 | | | -22.077 | | |

**Table 8. Comparison results for FOPID tuning using different methods over 100 different uncertainty scenarios for θ = 0.75.**

| Repeat No | Proposed algorithm | | | PSO | | | GWO | | | ALO | | |
|---|---|---|---|---|---|---|---|---|---|---|---|---|
| | SNR | Ave / Std | SE | SNR | Ave / Std | SE | SNR | Ave / Std | SE | SNR | Ave / Std | SE |
| 1 | -24.094 | -24.292 / 0.150 | 2448 | -23.774 | -23.198 / 1.675 | 2700 | -24.012 | -24.028 / 0424 | 2700 | -24.004 | -24.032 / 0.424 | 2700 |
| 2 | -24.523 | | | -19.045 | | | -24.262 | | | -23.300 | | |
| 3 | -24.295 | | | -24.050 | | | -24.607 | | | -24.253 | | |
| 4 | -24.395 | | | -24.077 | | | -23.667 | | | -24.608 | | |
| 5 | -24.471 | | | -21.273 | | | -23.075 | | | -24.596 | | |
| 6 | -24.276 | | | -24.416 | | | -24.504 | | | -23.998 | | |
| 7 | -24.136 | | | -24.385 | | | -23.845 | | | -24.064 | | |
| 8 | -24.444 | | | -22.591 | | | -23.855 | | | -24.191 | | |
| 9 | -24.136 | | | -24.384 | | | -24.258 | | | -24.004 | | |
| 10 | -24.153 | | | -23.986 | | | -24.198 | | | -23.300 | | |

optimization), the proposed algorithm shows the most robustness behavior (Std = 0.238) in comparison to the other three optimizers. The GWO, PSO, and ALO optimizers are the second, third, and fourth robust methods respectively.

From the results in Table 7, it is clear that the proposed algorithm provides competitive result in obtaining the lower SNR value with -22.112 compared to the lowest SNR value (-22.165) by GWO. The ALO and PSO are ranked third and fourth in obtaining lower objective function (SNR). In Table 7, it is readily apparent that the most robust method against randomness in the model is the proposed algorithm with Std = 0.280 compared to the other three optimizers. The next robust methods are GWO, ALO, and PSO optimizers respectively.

It is apparent from Table 8 that the lower SNR value (objective function) with -24.292 is obtained by the proposed algorithm compared to the other three optimizers. The ALO, GWO, and PSO optimizers are ranked second, third, and fourth respectively in obtaining lower SNR value. Regarding robustness against randomness in the stochastic optimization model, the proposed algorithm also shows the most robust behavior compared to the other three methods. The PSO optimizer shows a weak performance to obtain robustness (see Std statistics) in the current case study.

To perform further comparison of the proposed algorithm with three different optimizers including PSO, GWO, and ALO, we apply two common statistical tests including the *t*-test

**Table 9. *p*-values of the *t*-test and the Wilcoxon signed rank test for pairwise comparison of proposed algorithm with three common stochastic optimizers over 10 repetitions.**

| Optimizer | *t*-test | | | Wilcoxon signed rank test | | |
|---|---|---|---|---|---|---|
| | $\theta = 0.25$ | $\theta = 0.5$ | $\theta = 0.75$ | $\theta = 0.25$ | $\theta = 0.5$ | $\theta = 0.75$ |
| PSO | 0.339826 | 0.00999 | 0.04112 | 0.5 | 0.00778 | 0.00955 |
| GWO | 0.02200 | 0.36948 | 0.05283 | 0.00408 | 0.24815 | 0.07546 |
| ALO | 0.11113 | 0.09626 | 0.05497 | 0.16288 | 0.08681 | 0.04815 |

and the Wilcoxon signed ranks test [98]. Inside the field of inferential statistics, hypothesis testing [99] can be employed to draw inferences about one or more populations from given samples (results). In order to do that, two hypotheses, the null hypothesis $H_0$ and the alternative hypothesis $H_1$, are defined. The null hypothesis is a statement of no effect or no difference, whereas the alternative hypothesis represents the presence of an effect or a difference (in our case, significant differences between algorithms). Table 9 provides the *p*-values using the *t*-test and the Wilcoxon signed ranks test. These statistical test results computed for all the pairwise comparisons concerning the proposed algorithm compared with three optimizers of PSO, GWO, and ALO. In general, the Wilcoxon signed ranks test is safer than the *t*-test because it does not assume normal distributions. Also, the outliers (exceptionally good/bad performances of an algorithm in a few repetitions) have less effect on the Wilcoxon test than on the *t*-test [100]. As Table 9 states, the proposed algorithm shows a significant improvement over GWO for $\theta = 0.25$ with a level of significance $\alpha = 0.05$ regarding the *t*-test and with $\alpha = 0.01$ regarding the Wilcoxon test. For $\theta = 0.5$, the results for both types of test indicate an improvement of proposed algorithm over PSO with a level of significance $\alpha = 0.01$ and an improvement over ALO with $\alpha = 0.1$. The statistical comparison results for $\theta = 0.75$ using *t*-test shows an improvement of proposed algorithm over PSO with a level of significance $\alpha = 0.05$ and over GWO and ALO with $\alpha = 0.1$. The Wilcoxon test also for $\theta = 0.75$ indicates a significant improvement of proposed algorithm over PSO with a level of significance $\alpha = 0.01$, an improvement over GWO with $\alpha = 0.1$, and an improvement over ALO with $\alpha = 0.05$.

**4.5.2 Performance measure.** In many studies on optimization, the strength of an optimization technique is measured by comparing the final solution achieved by different algorithms [101, 102]. This approach only provides information about the quality of the results and neglects the speed of convergence which is a very important measure for expensive optimization problems. Comparing the convergence curves (number of function evaluations) is also one of the common benchmarking approaches [103]. A convergence curve provides good information about the final quality of the optimization results in terms of computational cost, even though it can be used to compare the performance of several algorithms only for one problem. Moré and Wild [104] have suggested performance measure for any pair (*p*, *s*) of problem *p* and solver *s* to analyze the performance of any optimization algorithm as below:

$$r_{p,s} = \frac{t_{p,s}}{\min\{t_{p,s'}\}}, \quad s, s' \in S \ and \ p \in P \tag{18}$$

where $P$ is a set of problems, $S$ is a set of solvers, and $t_{p,s}$ is the required number of the function evaluations for solver $s \in S$ to solves a particular problem $p \in P$. In Eq (18), larger values of $t_{p,s}$ indicate a worse performance. The convention $r_{p,s} = \infty$ is used when solver s fails to satisfy the convergence test on problem $p$. However, Eq (18) considers the required budget to solve the expensive optimization problem. In this study, inspired by [104], a new performance measure is used to consider two aspects of the algorithm's performance including the level of accuracy

and computational cost as below:

$$R_{p,s} = \left\{ \beta \frac{t_{p,s}}{\min\{t_{p,s'}\}} + (1 - \beta) \frac{l_{p,s}}{\min\{l_{p,s''}\}} \right\}, \quad s, s', s'' \in S \ and \ p \in P \tag{19}$$

where $l_{p,s}$ indicates the level of accuracy (i.e. lower objective function) for solver $s$ in an expensive problem $p$ and $\beta(0 \leq \beta \leq 1)$ is the weight scale. Note that the best solver for a particular problem $p$ obtains the lower bound $R_{p,s} = 1$. In $\beta = 1$, Eq (19) makes the same measure with suggested $r_{p,s}$ based on [104] in Eq (18). For $\beta = 0$, only the accuracy of solver in obtaining lower objective function is considered as a performance measure for comparing all the optimization methods. However, the computational cost (i.e. the number of the function evaluation) is not attended. Here, we measure the performance of the proposed algorithm and other solvers for both average and standard deviation of the optimal results obtained for 10 repetitions regarding the results presented in Tables 6–8. Moreover, we apply the Eq (19) when $\beta$ varies in [0,1] to obtain performance measures. The results are shown in Fig 13. This figure reports an appropriate performance ($R_{p,s}$) of the proposed algorithm for the stochastic FOPID tuning or robot manipulator compared to the other solvers when the budget of optimization for an expensive control system is limited to a small number of function evaluations. As mentioned before, in the current case we assume a maximum of 270 function evaluations.

## 5. Discussion

This paper followed two main purposes i) sketching a new framework of a stochastic control system for robot FOPID control under uncertainty in a CPS framework ii) developing a new optimization algorithm associated with such stochastic control systems. Regarding the second purpose, there are some rationales including:

- Most existing methods have been developed to apply in the deterministic control systems rather than the stochastic or random models [47, 54, 96]. Moreover, we firstly developed a new straightforward algorithm that can handle uncertainty for any stochastic behavior of control systems [35]. Notably, here we show the applicability of the proposed method for robust FOPID tuning of a robot manipulator in a CPS framework.

- Most practical control systems in CPS are complex in terms of dynamic mathematical sophistication or time-consuming simulation experiments [36, 47]. Therefore, we aimed to propose a new less-expensive method for complex black-box simulation models when a limited (small) number of input-output data needs be applied in the control system (i.e. the model is limited to a few numbers of simulation experiments or function evaluations).

- Besides optimal design (lower objective function) and robustness against sources of variability (uncertainty) with a small number of simulation experiments, we are also interested to perform the sensitivity analysis (bootstrapping) for the obtained results in the stochastic (random) control system. The proposed algorithm in this study can compute the two-sided confidence intervals for the obtained optimal results using the same set of data produced among optimization procedure and it doesn't need extra simulation experiments (function evaluations) for sensitivity analysis.

- As elucidated in the No Free Lunch (NFL) theorem by [105], a particular optimization method may show very promising results on a set of problems, but the same algorithm may show poor performance on a different set of problems [94]. This is also another motivation for conduction this study.

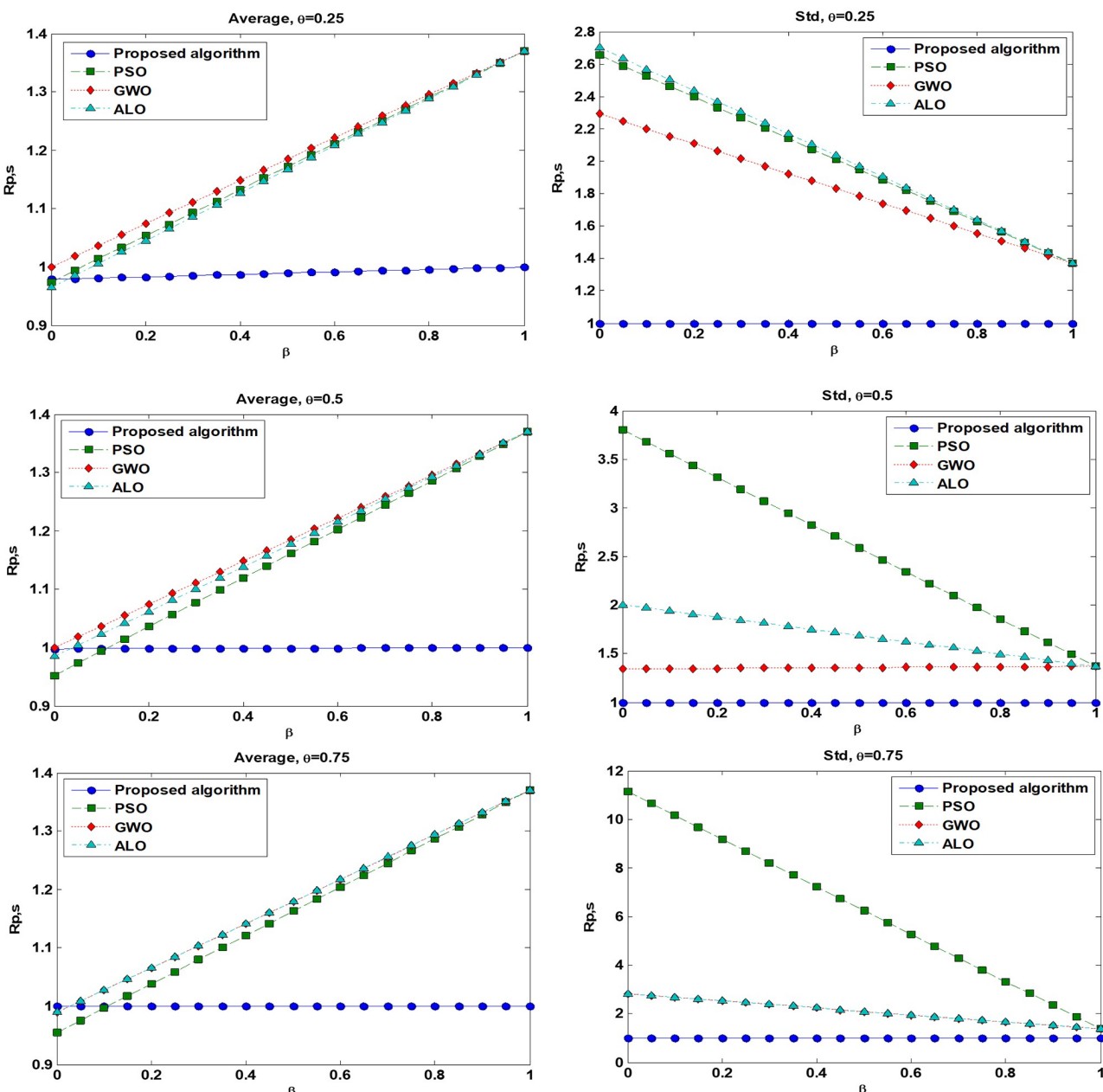

**Fig 13. The performance comparison of proposed algorithm with other three solvers in the literature for tuning of stochastic FOPID controller.** The performance criterion $R_{p,s}$ measured based on two terms, accuracy of solution (lower objective function) and number of function evaluations (computational cost), see Eq (19).

The proposed hybrid algorithm has been compared with the three most well-known optimizers (PSO, GWO, and ALO, see [49, 54, 96, 97] that are commonly used in control tunings. The comparison results have been provided in Section 4.5. The performance of the proposed algorithm can be evaluated by three factors as below:

- Level of accuracy (lower objective function): The proposed algorithm provides competitive results to obtain a lower objective function with three aforementioned optimizers.

- Robustness: The proposed algorithm shows a higher robustness behavior against randomness in the stochastic model of the control system.

- The number of simulation experiments (computational cost) that are required for optimization procedure.

The proposed hybrid algorithm has multi-disciplinary applications. In other words, the proposed algorithm can be applied in a variety of engineering design problems under the effect of uncertainty with expensive black-box simulation models. In this paper, we show the application of the proposed algorithm in the stochastic robot control system (robust tuning of FOPID controller for five-bar linkage robot manipulator).

The main limitations of the current study are as below:

- The proposed algorithm employed a Gaussian process (Kriging) surrogate to train the control system using space-filling sampling strategies. Therefore, the approximate errors could not be ignored when solving simulation-based optimization problems particularly with complex function and nonlinear structure. It is well known that GP is an ideal choice for smooth models. If the functions are non-smooth or noisy, it is likely that GP surrogate degrades rapidly and overfits due to its interpolating behavior. A challenge for optimization under restricted budgets will be to find the right degree of approximation (smoothing factor) from a limited number of samples [102].

- In this study, the proposed algorithm was evaluated in a stochastic control system with five design variables and two uncertain variables. The proposed algorithm should be evaluated more in other practical stochastic problems with a higher dimension and degree of uncertainty.

- Here, the two main weight-scale parameters including $\theta$ in Eq (6) and $\omega$ in Eq (13) mostly influenced the performance of the proposed algorithm in obtaining the robust optimal solution. The range of parameter $\theta$ was between zero and one ($0 \leq \theta \leq 1$) and any positive value could be assigned to the parameter $\omega$. In this study, we considered three different values for $\theta$ including $\theta = 0.25$, $\theta = 0.5$, and $\theta = 0.75$ and performed the optimization procedure separately. While the parameter $\omega$ was assumed to be a fix value $\omega = 3$ for all the values of $\theta$. This paper was limited to evaluate the effect of parameter $\theta$ only in obtaining the robust optimal solutions by the algorithm. However, more analysis is required to study the effect of both parameters of $\theta$ and $\omega$ at the same time and in the same platform of problem.

## 6. Conclusion

In this paper, a new CPS framework of fractional-order PID controller is developed by considering uncertainty in the control system. To optimize such a stochastic control system, a new hybrid surrogate/metaheuristic-based robust simulation-optimization algorithm is proposed that possesses the advantages of both GP surrogate in learning the behavior of the model for efficient global optimization and PSO metaheuristic in convergence searching of optimum results. We smooth the application of PSO using GP surrogate instead of the original simulation model to diminish computational cost due to a large number of fitness evaluations required for the global optimizer when used individually. Also, this simple modified algorithm is developed in such a way to handle computational complexity to obtain optimal and robust FOPID design in the CPS control system. In such a system, we also consider the conflict between multiple objective functions and uncertainty in the model. The proposed algorithm can analyze the sensitivity of the computed robust optimal results (i.e. it obtains two-sided

confidence intervals). Here, we apply this approach to the robust optimal control design of a five-bar linkage robot manipulator to depict the applicability and effectiveness of the proposed algorithm. Comparative simulation results reveal that the proposed hybrid GP/PSO-based robust efficient global optimization algorithm can effectively and robustly tune the parameters of the FOPID controllers. From an application point of view, the introduced technique is simple and fast and has a suitable control over the error and energy of a system and it can be easily implemented in real-world applications of CPS control systems. Future research may address the issues regarding the limitations of the current study mentioned in Section 5.

## Supporting information

**S1 File.**
(ZIP)

## Author Contributions

**Conceptualization:** Amir Parnianifard, Muhammad Ali Imran, Lunchakorn Wuttisittikulkij.

**Data curation:** Amir Parnianifard.

**Formal analysis:** Amir Parnianifard.

**Funding acquisition:** Ratchatin Chancharoen, Lunchakorn Wuttisittikulkij.

**Investigation:** Amir Parnianifard.

**Methodology:** Amir Parnianifard.

**Project administration:** Lunchakorn Wuttisittikulkij.

**Resources:** Amir Parnianifard.

**Software:** Amir Parnianifard.

**Supervision:** Ratchatin Chancharoen, Muhammad Ali Imran, Lunchakorn Wuttisittikulkij.

**Validation:** Amir Parnianifard, Ali Zemouche, Lunchakorn Wuttisittikulkij.

**Visualization:** Amir Parnianifard, Ali Zemouche.

**Writing – original draft:** Amir Parnianifard.

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
