## [Decision Letter · Decision Letter 0]

24 Sep 2020

PONE-D-20-25965

Robust Optimal Design of FOPID Controller for Five Bar Linkage Robot in a Cyber-Physical System: A New Simulation-Optimization Approach

PLOS ONE

Dear Dr. Parnianifard,

Thank you for submitting your manuscript to PLOS ONE. After careful consideration, we feel that it has merit but does not fully meet PLOS ONE’s publication criteria as it currently stands. Therefore, we invite you to submit a revised version of the manuscript that addresses the points raised during the review process.

We look forward to receiving your revised manuscript.

Kind regards,

Seyedali Mirjalili

Academic Editor

PLOS ONE

Journal Requirements:

Reviewers' comments:

Reviewer's Responses to Questions

**Comments to the Author**

1. Is the manuscript technically sound, and do the data support the conclusions?

Reviewer #1: Yes

Reviewer #2: Yes

2. Has the statistical analysis been performed appropriately and rigorously? 

Reviewer #1: Yes

Reviewer #2: N/A

3. Have the authors made all data underlying the findings in their manuscript fully available?

Reviewer #1: Yes

Reviewer #2: Yes

4. Is the manuscript presented in an intelligible fashion and written in standard English?

Reviewer #1: Yes

Reviewer #2: Yes

5. Review Comments to the Author

Reviewer #1: This is a good work, but a number of major and minor amendments are required as follows:

* As per NFL theorem, new algorithms are good for some set of problems. It is not clear which types of problem with what difficulties have been targeted by the authors when proposing the method.

* Potential applications of the proposed method should be discussed

* Some mathematical notations and Lemma presentations are not rigorous enough to correctly understand the contents of the paper. The authors are requested to recheck all the definition of variables and further clarify these equations.

* There is no justification of the method. Why for this problem area, please discuss. There are many other similar methods in the literature in this area, so such a justification is required.

* There is no statistical test to judge about the significance of the method’s results. Without such a statistical test, the conclusion cannot be supported.

* There is no discussion on the cost effectiveness of the proposed method. What is the computational complexity? What is the runtime? Please include such discussions. You can also use the big oh notation to show the computation complexity.

* To have an unbiased view in the paper, there should be some discussions on the limitations of the proposed method

* Analysis of the results is missing in the paper. There is a big gap between the results and conclusion. There should be the result analysis between these two sections. After comparing the methods, you have to be able to analyse the results and relate them to the structure of all algorithms. It would be interesting to have your thoughts on why the method works that way? Such analyses would be the core of your work where you prove your understanding of the reason behind the results. You can also link the findings to the hypotheses of the paper. Long story short, this paper requires a very deep analysis from different perspectives

* How do you ensure that the comparison between the proposed method and the comparative methods is fair?

* The proposed method might be sensitive to the values of its main controlling parameter. How did you tune the parameters?

Reviewer #2: This paper presents a hybridization of surrogate/metaheuristic optimization algorithm to obtain the optimal FOPID design in the CPS system. Overall, this paper is well written and organized. There are several observations listed as follows:

1. Page 2 line 43, please define CPS.

2. Page 4-5, it is suggested that the author to summarize the main contributions into 3-4 points.

3. Please provide citations for the utilized objective functions.

4. Please revise the Equation (10), the inertia weight was missing in the velocity update.

5. The algorithm was repeated for 5 runs/repetitions. Due to the stochastic of the metaheuristic algorithm, a minimum 10 runs should be performed.

6. The authors are suggested to provide the computational complexity of the proposed algorithm.

7. Please include the limitations of the research.

6. PLOS authors have the option to publish the peer review history of their article (what does this mean?). If published, this will include your full peer review and any attached files.

Reviewer #1: No

Reviewer #2: No

---

## [Author Response · Author response to Decision Letter 0]

22 Oct 2020

The authors would cordially thanks the respected associate editor and would appreciate esteemed reviewers for the useful comments and suggestions on the structure and contents of our manuscript. With respect to each comment, we have carefully modified the manuscript. The details of corrections are listed in the following point by point. The changes we made in the revised manuscript are marked in blue color. Politely, the manuscript has been resubmitted to the journal. We look forward to your positive response. 

Reviewer #1:

 As per NFL theorem, new algorithms are good for some set of problems. It is not clear which types of problem with what difficulties have been targeted by the authors when proposing the method. 

Response: The authors would be thankful for your insightful comment. The proposed algorithm is targeted to obtain robust optimal results in such expensive simulation-based optimization problems under the effect of uncertainty. The robust optimal design of fractional-order PID controllers particularly in the stochastic control systems is a challenging topic yet in the control engineering, see (Aghababa, 2016; Shah & Agashe, 2016; Verma et al., 2017). Here, we consider the real application of the proposed algorithm for intelligent robot control in a cyber-physical system (Blondin et al., 2019; Blondin & Pardalos, 2019). Two main difficulties considered in such stochastic control problems:

 Uncertainty existed in the control system, so robustness (insensitive results against changeability on the model’s parameters) needs to be obtained besides accuracy (lower objective function) in the optimal results. Most surrogate-based simulation-optimization methods have been developed to apply in the deterministic control systems, rather than the stochastic or random models. 

 A limited (small) number of simulation experiments (function evaluations) are allowed to perform in the robust optimization model. 

The proposed method is developed based on a hybrid algorithm employing the advantages of both Kriging surrogate and swarm optimizer. To tackle uncertainty in the model, we apply the Taguchi terminology, when we replace its statistical approach in the Taguchi method by design and analysis of simulation experiments (DACE) (Dellino & Meloni, 2015; Kleijnen, 2015). 

 Potential applications of the proposed method should be discussed. 

Response: Thank you so much for this helpful comment. The proposed hybrid algorithm has multi-disciplinary applications. It means that this algorithm can be used in a variety of engineering design problems under the effect of uncertainty with expensive black-box simulation models. In this paper, we show the application of the proposed algorithm in the robotics and control (robust tuning of FOPID controller for five-bar linkage robot manipulator). However, the application of the proposed algorithm needs to be shown in more engineering design problems among further research studies. 

 Some mathematical notations and Lemma presentations are not rigorous enough to correctly understand the contents of the paper. The authors are requested to recheck all the definition of variables and further clarify these equations. 

Response: The authors would cordially thank you for this comment. Concerning this comment, the paper has been rechecked accordingly. Besides, a nomenclature table has been added to the paper to reveal the main parameters and symbols used in the proposed algorithm. 

 There is no justification of the method. Why for this problem area, please discuss. There are many other similar methods in the literature in this area, so such a justification is required.

Response: The authors would appreciate this comment. The robust optimal design of fractional-order PID controllers particularly in the stochastic control system is a challenging topic yet in the control engineering, see (Aghababa, 2016; Shah & Agashe, 2016; Verma et al., 2017). This paper followed two main purposes i) sketching a new framework of a stochastic control system for robot FOPID control under uncertainty in a CPS framework ii) developing a new optimization algorithm associated with such stochastic control systems. Regarding the second purpose, there are some rationales including: 

 Most existing methods have been developed to apply in the deterministic control systems rather than the stochastic or random models (Aghababa, 2016; Shah & Agashe, 2016; Verma et al., 2017). Moreover, we firstly developed a new straightforward algorithm that can handle uncertainty for any stochastic behavior of control systems (Blondin et al., 2019). Notably, here we show the applicability of the proposed method for robust FOPID tuning of a robot manipulator in a CPS framework. 

 Most practical control systems in CPS are complex in terms of dynamic mathematical sophistication or time-consuming simulation experiments (Blondin & Pardalos, 2019; Shah & Agashe, 2016). Therefore, we aimed to propose a new less-expensive method for complex black-box simulation models when a limited (small) number of input-output data needs be applied in the control system (i.e. the model is limited to a few numbers of simulation experiments or function evaluations). 

 Besides optimal design (lower objective function) and robustness against sources of variability (uncertainty) with a small number of simulation experiments, we are also interested to perform the sensitivity analysis (bootstrapping) for the obtained results in the stochastic (random) control system. The proposed algorithm in this study can compute the two-sided confidence intervals for the obtained optimal results using the same set of data produced among optimization procedure and it doesn’t need extra simulation experiments (function evaluations) for sensitivity analysis. 

 As elucidated in the No Free Lunch (NFL) theorem by (Wolpert & Macready, 1997), a particular optimization method may show very promising results on a set of problems, but the same algorithm may show poor performance on a different set of problems (Mirjalili et al., 2014). This is also another motivation for conduction this study. 

The proposed hybrid algorithm has been compared with the three most well-known optimizers (PSO, GWO, and ALO, see (Aghababa, 2016; Pradhan et al., 2019; Tepljakov et al., 2018; Verma et al., 2017) that are commonly used in control tunings. The comparison results have been provided in Section 4.5. The performance of the proposed algorithm can be evaluated by three factors as below: 

 Level of accuracy (lower objective function): The proposed algorithm provides competitive results to obtain a lower objective function with three aforementioned optimizers.

 Robustness: The proposed algorithm shows a higher robustness behavior against randomness in the stochastic model of the control system. 

 The number of simulation experiments (computational cost) that are required for optimization procedure.

The main advantages of the current study can be highlighted as below:

 In this paper, we consider a new CPS framework of the control system for a five-bar linkage robot manipulator by the effect of uncertainty in the model. 

 A new algorithm for robust tuning of FOPID controller in the stochastic control system is proposed. The competitive results with three common optimizers in the literature show the effectiveness of the proposed algorithm in such stochastic models. 

 The proposed algorithm can analyze the sensitivity of obtained optimal results in such stochastic environments using the same collected data obtained among optimization procedure, and no need to run simulation more (i.e. does not increase the number of function evaluations for sensitivity analysis). 

 The proposed hybrid algorithm has multi-disciplinary applications. In other words, the proposed algorithm can be applied in a variety of engineering design problems under the effect of uncertainty with expensive black-box simulation models. In this paper, we show the application of the proposed algorithm in the stochastic robot control system (robust tuning of FOPID controller for five-bar linkage robot manipulator). 

 There is no statistical test to judge about the significance of the method’s results. Without such a statistical test, the conclusion cannot be supported. 

Response: Thank you for this helpful comment. With respect to this comment, a Subsection 4.5 has been added to the paper including the statistical and analytical results for 10 different repetitions of the proposed algorithm in comparison with three common optimizers in the optimization of robot control system (please also see Table 5, 6, and 7). 

 There is no discussion on the cost effectiveness of the proposed method. What is the computational complexity? What is the runtime? Please include such discussions. You can also use the big oh notation to show the computation complexity.

Response: The authors thank the reviewer for this helpful comment. The authors have provided a detailed discussion in Section 4.5 to compare more the obtained results with other most common state-of-the-art optimizers in solving stochastic robot control problems. In the mentioned section, the performance of the proposed algorithm is measured and compared in two terms, firstly the accuracy of the method (i.e. power to obtain lower objective function), and secondly a number of function evaluations (computational cost) that is required in the optimization procedure of stochastic model to obtain robust optimal results. However, in this paper, we consider the required number of function evaluations (number of simulation experiments). We aim to develop the proposed algorithm for such expensive simulation models when the small number of simulation experiments only can be used for optimization and sensitivity analysis of the model. Moreover, instead of computation time, the authors used a required number of simulation experiments (function evaluations) as well as accuracy in obtaining optimal results (lower objective function) for comparison (please see Section 4.5.1 and Section 4.5.2). 

 To have an unbiased view in the paper, there should be some discussions on the limitations of the proposed method. 

Response: Thank you for this helpful comment. The authors have been added the limitations of the current study in a discussion part (Section 5) accordingly as below: 

The main limitations of the current study are as below:

 The proposed algorithm employed a Gaussian process (Kriging) surrogate to train the control system using space-filling sampling strategies. Therefore, the approximate errors could not be ignored when solving simulation-based optimization problems particularly with complex function and nonlinear structure. It is well known that GP is an ideal choice for smooth models. If the functions are non-smooth or noisy, it is likely that GP surrogate degrades rapidly and overfits due to its interpolating behavior. A challenge for optimization under restricted budgets will be to find the right degree of approximation (smoothing factor) from a limited number of samples (Bagheri et al., 2017). 

 In this study, the proposed algorithm was evaluated in a stochastic control system with five design variables and two uncertain variables. The proposed algorithm should be evaluated more in other practical stochastic problems with a higher dimension and degree of uncertainty. 

 Here, the two main weight-scale parameters including θ in Eq.(6) and ω in Eq.(13) mostly influenced the performance of the proposed algorithm in obtaining the robust optimal solution. The range of parameter θ was between zero and one (0≤θ≤1) and any positive value could be assigned to the parameter ω. In this study, we considered three different values for θ including θ=0.25, θ=0.5, and θ=0.75 and performed the optimization procedure separately. While the parameter ω was assumed to be a fix value ω=3 for all the values of θ. This paper was limited to evaluate the effect of parameter θ only in obtaining the robust optimal solutions by the algorithm. However, more analysis is required to study the effect of both parameters of θ and ω at the same time and in the same platform of problem. 

 Analysis of the results is missing in the paper. There is a big gap between the results and conclusion. There should be the result analysis between these two sections. After comparing the methods, you have to be able to analyse the results and relate them to the structure of all algorithms. It would be interesting to have your thoughts on why the method works that way? Such analyses would be the core of your work where you prove your understanding of the reason behind the results. You can also link the findings to the hypotheses of the paper. Long story short, this paper requires a very deep analysis from different perspectives. 

Response: The authors would cordially thank you for this comment. Concerning this comment, the authors have been tried carefully to revise and modify the technical explanation over result analysis, comparing the methods, and the structure of the paper accordingly. Hope the modified structure and content of revised paper satisfy the reviewer's concern. 

 How do you ensure that the comparison between the proposed method and the comparative methods is fair? 

Response: Thank you for this insightful comment. In this paper, we aim to compare the performance of the proposed algorithm with the three most common solvers in stochastic control systems with three terms: i) lower objective function, ii) robustness, iii) a number of simulation experiments (computational cost). To provide a fair comparison, we apply the same simulation model in Matlab® Simulink environment, the same objective function (and same parameters adjustment), and particularly the same maximum number of simulation experiments (e.g. 270 function evaluations) for all solvers. Each solver repeated 10 times in the same platform and the same parameter adjustment to obtain statistical test results. The FOPID controller tuned by obtained optimal result with each solver and simulation model is rerun 100 times regarding 100 the same designed uncertainty scenarios to obtain robustness of each optimal point against changeability in uncertain factors. 

 The proposed method might be sensitive to the values of its main controlling parameter. How did you tune the parameters? 

Response: The authors would be thankful for your comment. Here, the two main weight-scale parameters including θ in Eq.(6) and ω in Eq.(13) mostly influence the performance of the proposed algorithm in obtaining the robust optimal solution. The range of parameter θ is between zero and one (0≤θ≤1), and any positive value can be assigned to the parameter ω. In this study, we consider three different values for θ including θ=0.25, θ=0.5, and θ=0.75 and perform the optimization procedure separately. While the parameter ω is assumed to be fix value ω=3 for all the values of θ. This paper is limited to only evaluate the effect of parameter θ in obtaining robust optimal solutions by the algorithm. However, more analysis is required to study the effect of both parameters of θ and ω at the same time and in the same platform of problem. 

Reviewer #2: 

 Page 2 line 43, please define CPS. 

Response: The authors would be thankful for your comment. The relevant sentence has been revised accordingly. 

 Page 4-5, it is suggested that the author to summarize the main contributions into 3-4 points.

Response: Thank you so much for this helpful comment. With respect to this comment, the main contributions of the paper have been revised accordingly. 

 Please provide citations for the utilized objective functions. 

Response: Thank you for this helpful comment. We have revised the relevant part accordingly to provide the referenced studies for our utilized objective functions. 

 Please revise the Equation (10), the inertia weight was missing in the velocity update.

Response: Appreciate the reviewer for this insightful comment. The relevant equation has been revised accordingly. 

 The algorithm was repeated for 5 runs/repetitions. Due to the stochastic of the metaheuristic algorithm, a minimum 10 runs should be performed.

Response: The authors thank the reviewer for this helpful comment. Concerning this comment, the algorithm has been repeated 10 times, and all relevant results in the paper, tables, and figures have been revised accordingly. 

 The authors are suggested to provide the computational complexity of the proposed algorithm 

Response: Thank you for this helpful comment. In this paper, we consider the required number of function evaluations (number of simulation experiments). We aim to develop the proposed algorithm for such expensive simulation models when the small number of simulation experiments only can be used for optimization and sensitivity analysis of the model. Moreover, instead of computation time, the authors used a required number of simulation experiments (function evaluations) as well as accuracy in obtaining optimal results (lower objective function) for comparison (please see Section 4.5.1 and Section 4.5.2). 

 Please include the limitations of the research.

Response: The authors would cordially thank you for this comment. The authors have been added the limitations of the current study in a discussion part (Section 5) accordingly as below: 

The main limitations of the current study are as below:

 The proposed algorithm employed a Gaussian process (Kriging) surrogate to train the control system using space-filling sampling strategies. Therefore, the approximate errors could not be ignored when solving simulation-based optimization problems particularly with complex function and nonlinear structure. It is well known that GP is an ideal choice for smooth models. If the functions are non-smooth or noisy, it is likely that GP surrogate degrades rapidly and overfits due to its interpolating behavior. A challenge for optimization under restricted budgets will be to find the right degree of approximation (smoothing factor) from a limited number of samples (Bagheri et al., 2017). 

 In this study, the proposed algorithm was evaluated in a stochastic control system with five design variables and two uncertain variables. The proposed algorithm should be evaluated more in other practical stochastic problems with a higher dimension and degree of uncertainty. 

 Here, the two main weight-scale parameters including θ in Eq.(6) and ω in Eq.(13) mostly influenced the performance of the proposed algorithm in obtaining the robust optimal solution. The range of parameter θ was between zero and one (0≤θ≤1) and any positive value could be assigned to the parameter ω. In this study, we considered three different values for θ including θ=0.25, θ=0.5, and θ=0.75 and performed the optimization procedure separately. While the parameter ω was assumed to be a fix value ω=3 for all the values of θ. This paper was limited to evaluate the effect of parameter θ only in obtaining the robust optimal solutions by the algorithm. However, more analysis is required to study the effect of both parameters of θ and ω at the same time and in the same platform of problem. 

References: 

Aghababa, M. P. (2016). Optimal design of fractional-order PID controller for five bar linkage robot using a new particle swarm optimization algorithm. Soft Computing, 20(10), 4055–4067. https://doi.org/10.1007/s00500-015-1741-2

Bagheri, S., Konen, W., Emmerich, M., & Bäck, T. (2017). Self-adjusting parameter control for surrogate-assisted constrained optimization under limited budgets. Applied Soft Computing Journal, 61, 377–393. https://doi.org/10.1016/j.asoc.2017.07.060

Blondin, M. J., & Pardalos, P. M. (2019). Computational Intelligence and Optimization Methods for Control Engineering (Vol. 150, Issue September). http://link.springer.com/10.1007/978-3-030-25446-9

Blondin, M. J., Sáez, J. S., & Pardalos, P. M. (2019). Control Engineering from Classical to Intelligent Control Theory—An Overview. In Computational Intelligence and Optimization Methods for Control Engineering (pp. 1–30). Springer.

Dellino, G., & Meloni, C. (2015). Uncertainty Management in Simulation- Optimization of Complex Systems. Springer.

Kleijnen, J. P. C. C. (2015). Design and analysis of simulation experiments (2nd). Springer. https://doi.org/10.1007/978-0-387-71813-2

Mirjalili, S., Mirjalili, S. M., & Lewis, A. (2014). Grey Wolf Optimizer. Advances in Engineering Software, 69, 46–61. https://doi.org/10.1016/j.advengsoft.2013.12.007

Pradhan, R., Majhi, S. K., Pradhan, J. K., & Pati, B. B. (2019). Optimal fractional order PID controller design using Ant Lion Optimizer. Ain Shams Engineering Journal, xxxx. https://doi.org/10.1016/j.asej.2019.10.005

Shah, P., & Agashe, S. (2016). Review of fractional PID controller. Mechatronics, 38(January 2020), 29–41. https://doi.org/10.1016/j.mechatronics.2016.06.005

Tepljakov, A., Alagoz, B. B., Yeroglu, C., Gonzalez, E., HosseinNia, S. H., & Petlenkov, E. (2018). FOPID Controllers and Their Industrial Applications: A Survey of Recent Results 1. IFAC-PapersOnLine, 51(4), 25–30. https://doi.org/10.1016/j.ifacol.2018.06.014

Verma, S. K., Yadav, S., & Nagar, S. K. (2017). Optimization of Fractional Order PID Controller Using Grey Wolf Optimizer. Journal of Control, Automation and Electrical Systems, 28(3), 314–322. https://doi.org/10.1007/s40313-017-0305-3

Wolpert, D. H., & Macready, W. G. (1997). No free lunch theorems for optimization. IEEE Transactions on Evolutionary Computation, 1(1), 67–82.

---

## [Decision Letter · Decision Letter 1]

26 Oct 2020

PONE-D-20-25965R1

Robust Optimal Design of FOPID Controller for Five Bar Linkage Robot in a Cyber-Physical System: A New Simulation-Optimization Approach

PLOS ONE

Dear Dr. Parnianifard,

Thank you for submitting your manuscript to PLOS ONE. After careful consideration, we feel that it has merit but does not fully meet PLOS ONE’s publication criteria as it currently stands. Therefore, we invite you to submit a revised version of the manuscript that addresses the points raised during the review process.

We look forward to receiving your revised manuscript.

Kind regards,

Seyedali Mirjalili

Academic Editor

PLOS ONE

Reviewers' comments:

Reviewer's Responses to Questions

**Comments to the Author**

1. If the authors have adequately addressed your comments raised in a previous round of review and you feel that this manuscript is now acceptable for publication, you may indicate that here to bypass the “Comments to the Author” section, enter your conflict of interest statement in the “Confidential to Editor” section, and submit your "Accept" recommendation.

Reviewer #1: (No Response)

Reviewer #2: (No Response)

2. Is the manuscript technically sound, and do the data support the conclusions?

Reviewer #1: (No Response)

Reviewer #2: Yes

3. Has the statistical analysis been performed appropriately and rigorously? 

Reviewer #1: (No Response)

Reviewer #2: No

4. Have the authors made all data underlying the findings in their manuscript fully available?

Reviewer #1: (No Response)

Reviewer #2: Yes

5. Is the manuscript presented in an intelligible fashion and written in standard English?

Reviewer #1: (No Response)

Reviewer #2: Yes

6. Review Comments to the Author

Reviewer #1: My comments have been addressed. . .

Reviewer #2: In the revised version, the paper has been improved. The authors have addressed most of my concerns. There are several minor observations as follows:

1. Please label the lines in Figures 7 and 8.

2. The results need to be supported by the statistical analysis such as t-test.

7. PLOS authors have the option to publish the peer review history of their article (what does this mean?). If published, this will include your full peer review and any attached files.

Reviewer #1: No

Reviewer #2: No

---

## [Author Response · Author response to Decision Letter 1]

3 Nov 2020

The authors would cordially thank the respected associate editor and would appreciate esteemed reviewers for the useful comments and suggestions on the structure and contents of our manuscript. With respect to each comment, we have carefully modified the manuscript. The details of corrections are listed in the following point by point. The changes we made in the revised manuscript are marked in blue color. Politely, the manuscript has been resubmitted to the journal. We look forward to your positive response. 

Reviewer #1: 

My comments have been addressed. 

Reviewer #2:

 Please label the lines in Figures 7 and 8.

Response: The authors would be thankful for your insightful comment. Both figures have been revised accordingly to include the labels. 

 The results need to be supported by the statistical analysis such as t-test.

Response: Thank you so much for your helpful comment. Two common statistical tests including the t-test and the Wilcoxon signed ranks test have been added in the new revised paper to support obtained results concerning the pairwise comparison of the proposed algorithm with three stochastic optimizers including PSO, GWO, and ALO. The statistical test results and the relevant discussion are provided in Table 8 and Section 4.5.1.

---

## [Decision Letter · Decision Letter 2]

6 Nov 2020

Robust Optimal Design of FOPID Controller for Five Bar Linkage Robot in a Cyber-Physical System: A New Simulation-Optimization Approach

PONE-D-20-25965R2

Dear Dr. Parnianifard,

We’re pleased to inform you that your manuscript has been judged scientifically suitable for publication and will be formally accepted for publication once it meets all outstanding technical requirements.

Kind regards,

Seyedali Mirjalili

Academic Editor

PLOS ONE

Additional Editor Comments (optional):

Reviewers' comments:

Reviewer's Responses to Questions

**Comments to the Author**

1. If the authors have adequately addressed your comments raised in a previous round of review and you feel that this manuscript is now acceptable for publication, you may indicate that here to bypass the “Comments to the Author” section, enter your conflict of interest statement in the “Confidential to Editor” section, and submit your "Accept" recommendation.

Reviewer #1: (No Response)

Reviewer #2: All comments have been addressed

2. Is the manuscript technically sound, and do the data support the conclusions?

Reviewer #1: (No Response)

Reviewer #2: (No Response)

3. Has the statistical analysis been performed appropriately and rigorously? 

Reviewer #1: (No Response)

Reviewer #2: (No Response)

4. Have the authors made all data underlying the findings in their manuscript fully available?

Reviewer #1: (No Response)

Reviewer #2: (No Response)

5. Is the manuscript presented in an intelligible fashion and written in standard English?

Reviewer #1: (No Response)

Reviewer #2: (No Response)

6. Review Comments to the Author

Reviewer #1: My comments have been addressed. .

Reviewer #2: In the revised paper, the authors have addressed all of my concerns.

7. PLOS authors have the option to publish the peer review history of their article (what does this mean?). If published, this will include your full peer review and any attached files.

Reviewer #1: No

Reviewer #2: No

---

## [Editor Report · Acceptance letter]

16 Nov 2020

PONE-D-20-25965R2 

Robust Optimal Design of FOPID Controller for Five Bar Linkage Robot in a Cyber-Physical System: A New Simulation-Optimization Approach 

Dear Dr. Parnianifard:

I'm pleased to inform you that your manuscript has been deemed suitable for publication in PLOS ONE. Congratulations! Your manuscript is now with our production department. 

Kind regards, 

on behalf of

Prof. Seyedali Mirjalili 

Academic Editor

PLOS ONE